# Transparent radiative cooling cover window for flexible and foldable electronic displays

Kang Won Lee[1,2], Jonghun Yi[1,2], Min Ku Kim[1] & Dong Rip Kim [1] ✉

Transparent radiative cooling holds the promise to efficiently manage thermal conditions in various electronic devices without additional energy consumption. Radiative cooling cover windows designed for foldable and flexible displays could enhance cooling capacities in the ubiquitous deployment of flexible electronics in outdoor environments. However, previous demonstrations have not met the optical, mechanical, and moisture-impermeable criteria for such cover windows. Herein, we report transparent radiative cooling metamaterials with a thickness of 50 microns as a cover window of foldable and flexible displays by rational design and synthesis of embedding optically-modulating microstructures in clear polyimide. The resulting outcome not only includes excellent light emission in the atmospheric window under the secured optical transparency but also provides enhanced mechanical and moisture-impermeable properties to surpass the demands of target applications. Our metamaterials not only substantially mitigate the temperature rise in heat-generating devices exposed to solar irradiance but also enhance the thermal management of devices in dark conditions. The light output performance of light-emitting diodes in displays on which the metamaterials are deployed is greatly enhanced by suppressing the performance deterioration associated with thermalization.

In contrast to conventional flat displays, foldable and flexible displays offer a range of form factors for implementing flexible electronics[1–3]. As the outermost layer of these displays, developing a foldable and flexible cover window is crucial to replace the conventional rigid cover glass[1,3–5]. To fulfill the requirements of the cover window for foldable and flexible displays, the desired characteristics include optimal optical properties for precise image delivery, robust mechanical properties for display protection from mechanical damage, and moisture-impermeable properties to protect the underlying devices from high humidity[1,3,5–7]. Ultrathin glass (UTG) and clear polyimide (c-PI) are excellent base materials for foldable and flexible cover windows[1,3,4]. While the fragility of UTG hinders its broad applications in foldable and flexible displays, c-PI-based polymeric cover windows have been further developed by fabricating composites[2,3,8] or modifying a base material[1,9] to improve the desired functionalities, including an increased elastic modulus[3,4,8,9] and enhanced moisture-impermeable properties[6,7,10] under secured optical transparency.

High-performance electronic devices with high-brightness displays require effective heat dissipation of internal devices to secure their lifetime[11–16]. The ubiquitous deployment of flexible electronics in indoor and outdoor environments has led to challenges in conventional thermal management methods due to excess heat from internal devices and external solar irradiance. Previous efforts on the effective thermal management of electronic devices with high-brightness displays focused on effective heat dissipation through the backsides or cooling channels of electronic devices by developing composites with highly thermally conductive materials[11,12,14,17] or by arranging large-sized heat spreaders or heat sinks for enhanced cooling capacities[13,18]. Despite their successful demonstrations, developing diverse strategies for effective thermal management is desired for electronic devices

[1]School of Mechanical Engineering, Hanyang University, Seoul 04763, South Korea. [2]These authors contributed equally: Kang Won Lee, Jonghun Yi.
✉e-mail: dongrip@hanyang.ac.kr

with confined spaces to engage in additional heat supply from external solar irradiance. From this perspective, there is limited focus on developing thermal management strategies through cover windows.

Radiative cooling, a promising passive cooling technology that can dissipate heat to outer space through the atmospheric window (8–13 μm) with no additional energy input, has been highlighted because of its potential to lower or maintain target temperatures, thereby alleviating cooling loads[15,19–21]. One representative strategy is to realize opaque radiative cooling materials to achieve sub-ambient cooling by effectively reflecting the solar spectrum (0.3–2.5 μm) and increasing the emission of the atmospheric window[20,22,23]. However, opaque radiative cooling materials are inadequate for applications that require optical transparency, such as windows and solar cells. As such, the development of optically transparent radiative cooling materials that effectively suppress the temperature rise of target devices upon solar irradiation has recently been suggested[19,24–29]. For example, the fabrication of transparent silica photonic crystals effectively suppresses the temperature rise of the underlying substrate by up to 13 °C in California, USA[29]. A silica micro-grating photonic cooler exhibited excellent visible transmittance (-90%) and light emission (-90%) in the atmospheric window, reducing the temperature rise of the commercial silicon solar cell by 3.6 °C under solar irradiance of 830–990 W/m$^2$ [27]. To achieve the scalability and flexibility of optically transparent radiative cooling materials, randomly dispersed silica particles in polymethylpentene were synthesized to realize translucent radiative cooling metamaterial films with light emission of -93% in the atmospheric window[26]. A transparent radiative cooling composite thin-film in which silica nanospheres were randomly distributed in polymenthylpentene (TPX) achieved transmittance of -90% in the wavelengths of 0.3–1 μm and light emission of 85% in the mid-infrared wavelengths of 8–20 μm, thereby leading to 5 °C reduction of solar cell temperatures under direct sunlight[24]. The fabrication of transparent bamboo-derived composites with epoxy infiltration demonstrated a visible-light transmittance of -80% and light emission of -95% at wavelengths of 2.5–20 μm[28]. Although the increased content of silica (emissivity of -0.94[23]) microstructures in the polymer matrix (e.g., polydimethylsiloxane (PDMS), emissivity of -0.92[30]) enhances light emission in the atmospheric window, it inevitably sacrifices visible clarity. To resolve this issue, the inclusion of silica aerogel microparticles mixed with an optical modulator (n-hexadecane) in a PDMS film enabled visible clarity owing to refractive index matching between the optical modulator and PDMS while securing sufficient silica content for effective radiative cooling[19]. It achieved visible transparency of >91% and light emission of >98% in the atmospheric window, leading to the temperature rise suppression of silicon substrates by up to 8.5 °C in Seoul, South Korea (solar intensity of -920 W/m$^2$ and relative humidities of 30–45%).

When the radiative cooling functionality is equipped with cover windows on foldable and flexible displays, it has the potential to act as an effective thermal management platform to manage the excess heat supply from internal devices and external solar irradiance. Although the cooling performance of transparent radiative cooling materials has been demonstrated, their poor mechanical properties, such as low elastic modulus and low scratch resistance, remain inadequate for fulfilling the requirements of cover windows for foldable and flexible displays. Hence, material formulations must be further developed to fulfill the desired optical, mechanical, and moist-impermeable properties of the cover window for foldable and flexible displays. Particularly, c-PI is a promising candidate because of its excellent optical and mechanical properties and moisture impermeability.

In this work, we demonstrate optically transparent radiative cooling metamaterials based on c-PIs for the cover window of foldable and flexible displays. While the mechanical compliance required for foldable and flexible cover windows limits the total thickness to approximately 50 μm[1,5], c-PI has low emissivity in the atmospheric window[31], which makes it challenging to endow radiative cooling

functionality to c-PI. To overcome the limitation, the optically modulating microstructures (polymethyl methacrylate (PMMA)-infiltrated silica (SiO$_2$) aerogel microparticles, hereafter PMMA-SiO$_2$ microstructures) are introduced by homogeneously dispersing those in c-PI. The synthesized transparent radiative cooling metamaterials exhibit a high visible transmission of 85.5% (i.e., 97.1% of the bare c-PI visible transmission) in the 400–800 nm wavelengths with a controlled haze from 25% to 64%. In addition, including PMMA-SiO$_2$ microstructures in c-PI achieves a 2.2 times higher elastic modulus and 0.6 times lower water vapor transmission rates than bare c-PI, which is favorable for the cover window of foldable and flexible displays. Finally, the synthesized 50 μm thick metamaterials significantly increase the light emission of c-PI from 60.2% to 94.6% in the atmospheric window, effectively suppressing the temperature rise of the displays in both outdoor and indoor environments. We compare our transparent metamaterials with the previously demonstrated transparent radiative cooling materials in terms of their key features and radiative cooling performance in Supplementary Table 1. The cover window materials of foldable and flexible displays should have high elastic modulus, enabling the excellent elastic recovery and rebound resilience[3]. While transparent radiative cooling materials based on SiO$_2$ are stiff and brittle for the cover windows of foldable displays, PDMS-based transparent radiative cooling materials do not meet the folding requirement due to the considerably low elastic modulus. TPX, Silk fibroin and cellulose-based transparent radiative cooling materials have poor moisture impermeability. This study demonstrates the excellent radiative cooling performance of metamaterials which can address the requirements of the cover windows of foldable and flexible displays.

## Results
### Synthesis of transparent metamaterials for display cover windows
Figure 1a illustrates the working principles of the transparent radiative cooling metamaterials used as cover windows for foldable and flexible displays. Our metamaterials were synthesized by randomly distributing 6-24 wt% PMMA-SiO$_2$ microstructures in c-PI. First, the phonon-polariton resonance of SiO$_2$ at a wavelength of 9.7 μm plays an important role in emitting light in the atmospheric window, while shaping SiO$_2$ into microstructures enables Mie scattering to further enhance the light emission in those wavelengths[23,32]. Although the selective emissivity in the atmospheric window is preferred to realize sub-ambient radiative cooling[20], the broadband radiator is desirable to suppress the temperature rise of the device which has higher temperature than ambient by emitting more heat than the incoming atmospheric radiation, thereby increasing net cooling power[33]. Since SiO$_2$ microstructures have different extinction coefficients in terms of their sizes[23,26], the broad distribution of SiO$_2$ microstructures contributes to increased light emission at broadband wavelengths, suppressing the temperature increase[23,26]. Hence, SiO$_2$ aerogel microparticles sized 4–10 μm, where SiO$_2$ nanospheres with a diameter of -20 nm form a chain-like network[34] (Supplementary Fig. 1), are employed in metamaterials. Second, the excellent visible transparency of the radiative cooling metamaterials under enhanced light emission in the atmospheric window is enabled by synthesizing composites of PMMA-SiO$_2$ microstructures with full infiltration of PMMA into the SiO$_2$ aerogel microparticles by replacing the air inside the SiO$_2$ aerogel microparticles with PMMA. Specifically, the similar refractive indices of PMMA (1.49) and c-PI (1.50) and the low extinction coefficients of PMMA (-0) at visible wavelengths are exploited to suppress light reflection and absorption by the microstructures. Full infiltration of PMMA into the SiO$_2$ aerogel microparticles generates the effect of making it look as if SiO$_2$ nanospheres are arranged in microstructured shapes within the c-PI. Full infiltration of PMMA into the SiO$_2$ aerogel microparticles is possible owing to the low viscosity (-0.01 g/cm·s) of PMMA diluted in anisole and low surface energy (-31 mN/m) and

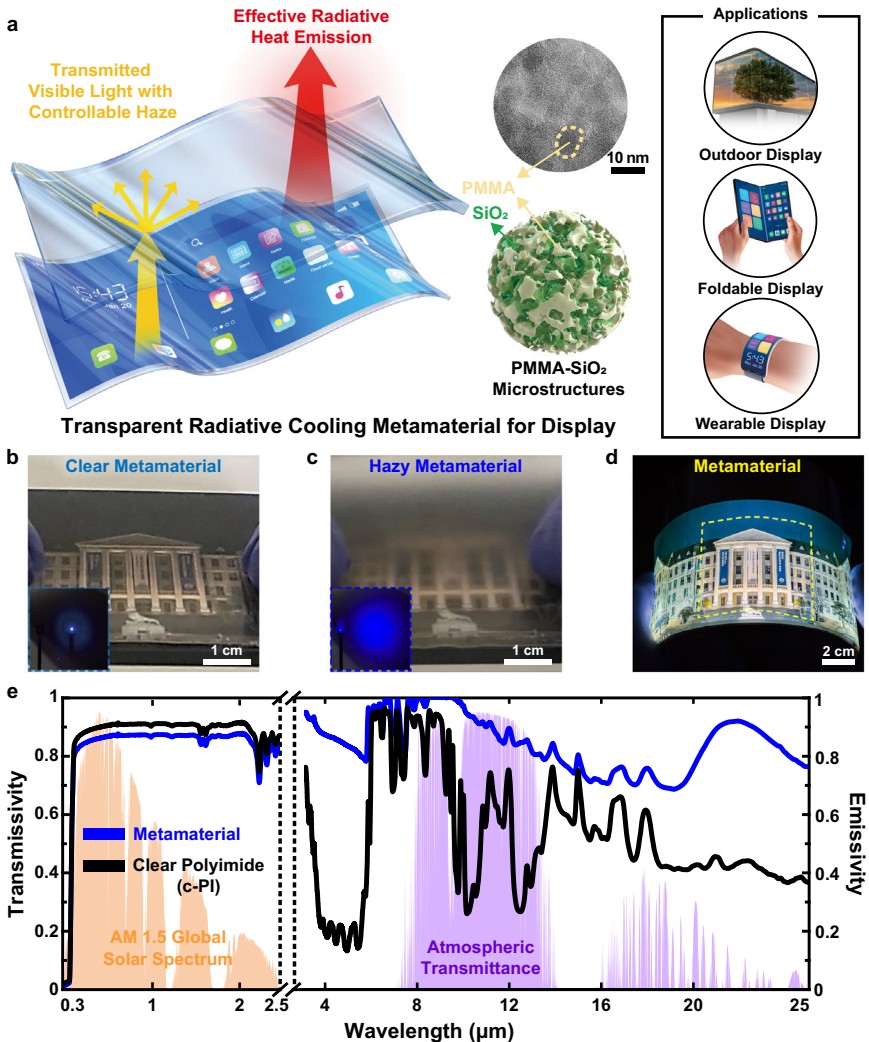

**Fig. 1 | Transparent radiative cooling materials for foldable cover window.**
**a** Schematical illustration of working principle and applications. PMMA-SiO$_2$ microstructures, an emissive material, provide radiative cooling functionality with minimal sacrifice of optical transparency. The upper schematic of PMMA-SiO$_2$ microstructures represents the transmission electron microscopy (TEM) image of PMMA-SiO$_2$ microstructures. Photographs to show the metamaterials with controllability of optical haze by **b** clear images for visible clarity and **c** hazy images for anti-glare functionality and **d** mechanical flexibilities attached on a flexible display. **e** Measured transmittance in the AM1.5 G solar spectrum and emissivity in the atmospheric window of the metamaterial and c-PI with the identical thickness of 50 μm. All the samples in Fig. 1 are 24 wt% PMMA-SiO$_2$ microstructures. Source data are provided as a Source Data file.

similar solubility of PMMA with c-PI. The detailed parameters are listed in Supplementary Table 2. It should be noted that the bending vibrations of the C-H, C = O, and C-O groups in PMMA can further improve the light emission in the atmospheric window[35]. Increasing the contents of PMMA-SiO$_2$ microstructures in c-PI accompanies the enhanced visible light scattering, thereby controlling optical haze for anti-glare effects in outdoor applications. Consequently, clear and hazy metamaterials were successfully synthesized in uniform manner over large areas, as shown in Fig. 1b–d. Our radiative cooling materials (24 wt% PMMA-SiO$_2$ microstructures) with 50 μm film thickness exhibit 85.5% transmission of visible light (400–800 nm) while demonstrating 94.6% and 89.2% light emission in the atmospheric window (8–13 μm) and mid-infrared wavelengths (2.5–25 μm), respectively.

## Optical, mechanical, and moisture-impermeable properties of transparent metamaterials

To fulfill the requirements of cover windows for foldable and flexible displays, the optical, mechanical, and moisture-impermeable properties of the transparent radiative cooling metamaterials were investigated in terms of the content of the PMMA-SiO$_2$ microstructures (0–24 wt%) under an identical film thickness of 50 μm, as shown in Fig. 2. While the PMMA-SiO$_2$ microstructures embedded in c-PI secure sufficient transmission of visible light owing to the refractive index matching of PMMA and c-PI (Fig. 2a), increasing the contents of the PMMA-SiO$_2$ microstructures from 0 to 24 wt% effectively increases the optical haze from 0.1 to 0.64 by promoting light scattering, enabling broad applications of displays from clear visualization to anti-glare effects (Fig. 2b). Despite the refractive index matching of PMMA and c-PI, they still have a slight difference in the refractive index at visible wavelengths, and a large degree of inclusion of PMMA-SiO$_2$ microstructures induces light scattering effects to increase the optical haze with minimal sacrifice of the total transmittance. The commercialized anti-glare films exhibit the haze factor of 0.1–0.4[36], but the anti-glare films with higher optical haze (0.5–0.9[37–39]) are desirable to enhance light coupling efficiency of light-emitting diode systems for displays operating in bright environments[37]. The capability to tune the haze factor of the metamaterials by varying the contents of PMMA-SiO$_2$ microstructure offers the potential for enhanced customization and optimization in the specific applications. It should be noted that the inclusion of the air-SiO$_2$ microstructures to c-PI not only considerably

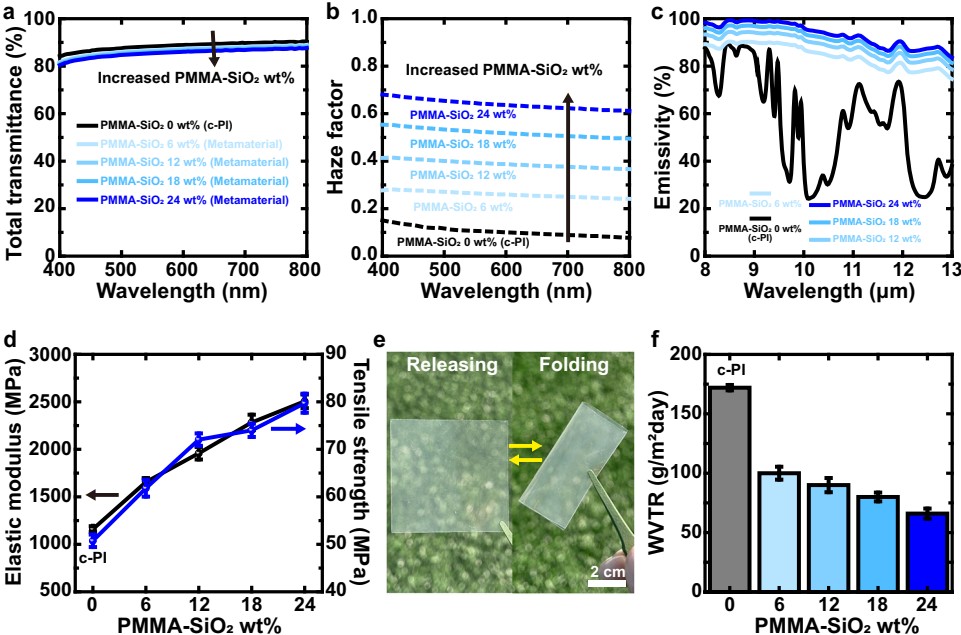

**Fig. 2 | Metamaterial with essential properties as a foldable and flexible display cover window.** Comparisons of the measured optical properties of metamaterials in terms of PMMA-SiO$_2$ microstructures contents (0–24 wt%). **a** Total transmittance, **b** haze factor in visible wavelengths (400–800 nm), and **c** emissivity in the atmospheric window (8–13 μm) of metamaterials are obtained with different wt% of PMMA-SiO$_2$ microstructures. **d** Elastic modulus and tensile strength of metamaterials are measured, while increasing the contents of PMMA-SiO$_2$ microstructures to confirm the mechanical properties of the cover window. **e** Photographs to illustrate the foldability of metamaterial under repeating bending cycles. **f** Water vapor transmission rate (WVTR) values which indicates moisture-impermeable properties of metamaterials with respect to the contents of PMMA-SiO$_2$ microstructures in c-PI at 25 °C and 100% relative humidity. Error bars in **d** and **f** indicate variations in measurements of the samples, displaying the mean and standard distribution (**d**, $n = 15$ and **f**, $n = 30$). Source data are provided as a Source Data file.

sacrifices the transmission in the visible wavelengths, but also greatly increases the haze factor even with their small inclusion due to their light scattering effects (Supplementary Fig. 2 and Supplementary Fig. 3). Adding the PMMA-SiO$_2$ microstructures to c-PI considerably increases light emission in the atmospheric window. While 50 μm thick c-PI has light emission of ~60% in the atmospheric window, only 6 wt% PMMA-SiO$_2$ microstructures greatly increase light emission to ~86% in the atmospheric window, which is further increased to ~95% with 24 wt% PMMA-SiO$_2$ microstructures to show the excellent heat dissipating capabilities by radiative cooling under the identical film thicknesses (Fig. 2c). Increasing the film thickness can enhance the light emission in the atmospheric window by sacrificing visible light transmission (Supplementary Fig. 4). Excellent light emission properties in an atmospheric window with a thin film (thickness of 50 μm) can improve the effective heat dissipation of radiative cooling by reducing the thermal insulating effects of heat conduction[40]. To further increase the broadband radiation of our metamaterials, the chemical modification of emissive materials[41] or the microstructure optimization with respect to size[23] and arrangement[25] can be considered.

While the cover window requires high mechanical strength to protect the display from mechanical damage, it also necessitates a foldable property that does not induce fragility upon repeated folding and release[1,3,42]. For foldable displays, the cover windows are preferred to endure the bending with a radius of 1 mm without degradation[43]. To accomplish the goal, the desired mechanical requirements are elastic modulus of >1 GPa, tensile strengths of >50 MPa, and elongation at break of >2.5% (film thickness of 50 μm)[3,9,44]. It should be noted that high elastic modulus enables the excellent elastic recovery and rebound resilience. Otherwise, the repeated bending cycles greatly generate the wrinkle formation on the surfaces[45]. Introducing zirconia nanoparticles and cellulose nanocrystals to c-PI increased the elastic modulus from 1.43 GPa to 2.22 GPa, the tensile strengths from 66 MPa to 82 MPa, elongation at break of >4%

for foldable and flexible cover windows[3]. The high elastic moduli (~3.1 and ~71.7 GPa)[46–49] and tensile strengths (~70 and ~360 MPa)[46–48,50,51] of PMMA and SiO$_2$ enhance the mechanical properties of c-PI (an elastic modulus of 1.16 GPa and a tensile strength of 50.6 MPa) favorable to the cover window of foldable and flexible displays upon the addition of PMMA-SiO$_2$ microstructures (Fig. 2d, e). As a result, the addition of 24 wt% PMMA-SiO$_2$ microstructures to c-PI achieves 2.2 times and 1.6 times higher elastic modulus (2.51 GPa) and tensile strength (79.8 MPa) higher than those of c-PI, while exhibiting elongation at break of >3.7% (Fig. 2d and Supplementary Fig. 5). Our radiative cooling metamaterials with 24 wt% PMMA-SiO$_2$ microstructures demonstrate excellent foldable properties upon repeated folding and releasing processes, as shown in Fig. 2e. We carried out the bending cycle test for 10,000 bending times[52]. Specifically, 50 μm thick metamaterial film (24 wt% PMMA-SiO$_2$ microstructures) was bent and released with a bending radius of 1 mm as shown in Supplementary Fig. 6. After the bending cycle test, the tested film was attached on top of the LED chip to measure light output power, confirming the consistent performance of the metamaterials upon the repeated folding process (Supplementary Fig. 6a). In addition, SEM images reveal no distinct morphology changes, including wrinkle formation, on the metamaterial surfaces upon the repeated folding process (Supplementary Fig. 6b), showing the excellent foldable properties under repetitive mechanical stress.

It should be noted that the metamaterials with 28 wt% PMMA-SiO$_2$ microstructures show elongation at a break of 1.8% under tensile tests (Supplementary Fig. 5a). Moreover, although PDMS is a representative thermal emitter, it does not provide adequate capabilities as a cover window to enable display protection from mechanical damage, owing to its low mechanical strength (Supplementary Fig. 5b, c). The bending cycle test on the PDMS film with a bending radius of 1 mm for 10,000 bending times greatly generates the wrinkle formation on the PDMS surface (Supplementary Fig. 7) which is consistent to the previous report[45]. In addition, our transparent metamaterials greatly enhance

the moisture-impermeable properties of c-PI, owing to the increased diffusion path of moisture due to PMMA-SiO₂ microstructures (Fig. 2f). The higher content of PMMA-SiO₂ microstructures in the metamaterials leads to longer traveling paths of moisture in the metamaterials, effectively reducing the water vapor transmission rates (WVTR) from ~172 g/m²·day for c-PI to ~66 g/m²·day for the metamaterials with 24 wt % PMMA-SiO₂ microstructures. In addition, a decreased dielectric constant or increased film thickness of the polymeric cover-window materials can deteriorate the touch sensitivity of capacitive touch display[3,53,54]. From this perspective, an identical metamaterial film thickness with higher dielectric constants of emissive materials (PMMA and SiO₂) compared to c-PI[3,55,56] can enhance the touch sensitivity of polymeric cover windows. Therefore, the inclusion of PMMA-SiO₂ microstructures in our transparent radiative cooling metamaterials benefits the favorable mechanical and moisture-impermeable properties of the cover windows of foldable and flexible displays while securing high degrees of transmission in the visible wavelengths and light emission in the atmospheric window. Our strategy to synthesize the metamaterials with c-PI can be applied to other materials (e.g., PET, PVDF, etc.) by replacing the optically-modulating material (PMMA) with others to consider refractive index matching, low viscosity, low surface energy, and similar solubility. For the successful

implementation of our strategy to the cover windows of diverse foldable and flexible displays, considering touch sensitivity and thermal stability is also desirable in addition to optical transparency, mechanical property (foldability), and moisture impermeability. From that point of view, we summarize the characteristics of various pristine materials in Supplementary Table 3. For the commercial products, clear polyimide (c-PI) and ultra thin glass (UTG) have been widely studied as the candidate for foldable and flexible displays[3]. UTG is mainly composed of SiO₂ which has an emissivity of ~0.75[29] in the atmospheric window. Although UTG is flexible, it has weak foldability. It should be noted that for foldable displays, the cover windows are preferred to endure the bending with a radius of 1 mm without degradation[43].

## Radiative cooling characteristics in indoor and outdoor conditions

The radiative cooling effects of the synthesized metamaterials were investigated by real-time monitoring of the temperature increase in the simulated display to which the metamaterial film was attached (Fig. 3). Specifically, metamaterial films with different thicknesses were attached as cover windows to simulated displays with heat generation of 100 W/m² to mimic the heat emission of smartphones[57,58]. To

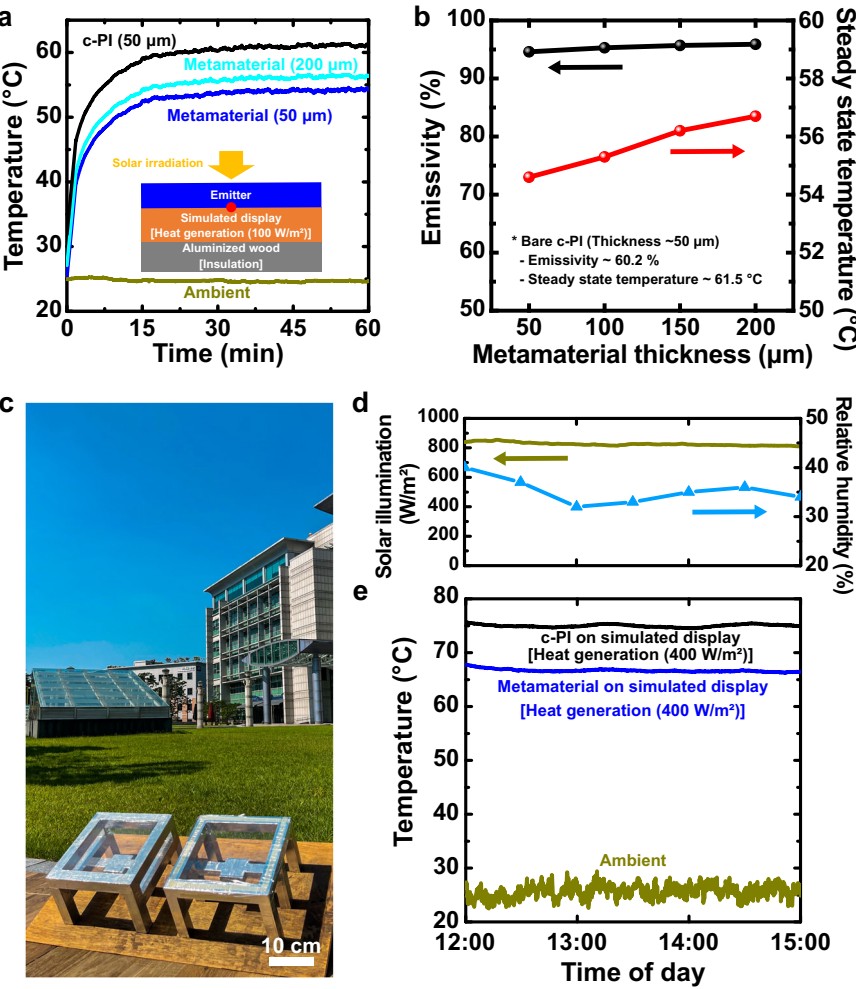

**Fig. 3 | Radiative cooling performance of the metamaterials (24 wt% PMMA-SiO₂ microstructures) on top of the heater, which is simulated display, in indoor (a–b) and outdoor environments (c–e). a** Measured heater (100 W/m²) temperature covered by c-PI (thickness of 50 μm) and metamaterials (thicknesses of 50 μm and 200 μm). The inset shows the experimental setup. **b** Emissivity in the atmospheric window and steady-state temperature in indoor (AM 1.5 G solar spectrum, 1,000 W/m²) conditions according to metamaterial thickness. **c** Photograph, **d** solar illumination, relative humidity, and **e** temperature variations of ambient, metamaterial on a heater (400 W/m²) and a control sample with c-PI on a heater (400 W/m²) at outdoor condition under solar irradiation in Seoul, South Korea. Source data are provided as a Source Data file.

quantify the radiative cooling effects of the metamaterials, the samples were placed on an aluminized wood frame to prevent heat gain, which was covered with low-density polyethylene (LDPE) film to minimize heat convection effects. Figure 3a, b show the effective suppression of temperature increase in the simulated devices by the metamaterial cover windows. The 50 μm thick transparent radiative cooling metamaterials (a steady-state temperature of 54.6 °C) with 24 wt% PMMA-SiO$_2$ microstructures suppress the temperature rise of the simulated display by 6.9 °C in indoor solar illumination of AM 1.5 Global (1000 W/m$^2$) at an ambient temperature of 24 °C and relative humidity of 30%, compared to the c-PI with the identical thickness (a steady-state temperature of 61.5 °C). It should be noted that the increased contents of PMMA-SiO$_2$ microstructures in the 50 μm thick metamaterials from 6 wt% to 24 wt% enhance the suppression of the temperature rise of the simulated display from 3.6 °C to 6.9 °C (Supplementary Fig. 8). As a control experiment, the radiative cooling performance of the 24 wt% air-SiO$_2$ microstructures in c-PI under the identical film thickness was characterized on the simulated display with heat generation of 100 W/m$^2$. As shown in Supplementary Fig. 9, the air-SiO$_2$ microstructure sample exhibits the suppression of the temperature rise by 4.9 °C, showing the better radiative cooling performance of the metamaterial. Although the increased thickness of the metamaterials from 50 μm to 200 μm further slightly increases the light emission from 94.6% to 95.9% in the atmospheric window, the steady-state temperatures of the simulated displays instead increase from 54.6 °C to 56.7 °C due to the increased thermal resistance of heat conduction (Fig. 3b).

The radiative cooling performance of the transparent metamaterials was experimentally confirmed under outdoor conditions (Seoul, South Korea) (Fig. 3c–e). To compare the temperature responses under identical solar illuminated conditions, two aluminized wood frames covered with LDPE films, which were tilted by 30° to receive direct sunlight around noon, were placed. Simulated displays with a heat generation of 400 W/m$^2$ were employed to simulate the heat emissions of outdoor digital displays[13,57]. The transparent metamaterial on top of the simulated display considerably suppresses the temperature rise of the simulated display by 8.3 °C, compared to the c-PI on top of the simulated display under solar irradiance of 850 W/m$^2$ at an ambient temperature of 23–29 °C and a relative humidity of 30–40%. Notably, the temperature behaviors of the simulated displays alone are similar to each other under illuminated and dark conditions (Supplementary Fig. 10), confirming that the suppression of the temperature rise of the simulated display originated from the radiative cooling metamaterials. The radiative cooling effects of the metamaterials on top of the simulated display with a heat generation of 100 W/m$^2$ are also confirmed in outdoor conditions, which reveals the suppression of the temperature rise of the simulated display by 5.1 °C, compared to the c-PI on top of the simulated display (Supplementary Fig. 11). This implies that the radiative cooling of our transparent metamaterials can be effective for diverse electronic devices with a wide range of heat generation. Furthermore, the theoretical estimation of our radiative cooling metamaterials[19,25] was carried out for indoor and outdoor environments (Supplementary Fig. 12). For indoor condition, the re-emission power from the surrounding should be high due to the existence of the ceiling and walls. When the emissivity of the surrounding is set as 1.0 in all the infrared region (2.5–25 μm) for the indoor environment to consider those as an enclosure, the net cooling power of the metamaterial film should be < 0 W/m$^2$ under direct solar light at ambient temperature. Nonetheless, the increased temperature difference between the metamaterial and the surrounding greatly increases the net cooling power of the metamaterial, indicating the effective suppression of the temperature rise of the target devices by integrating the metamaterial film. In outdoor environments, the theoretical estimation indicates the excellent net cooling power of ~109 W/m$^2$ under direct solar light at ambient temperature. This is comparable to the representative transparent thermal

emitter of randomly dispersed silica particles in polymethylpentene with a net cooling power of ~93 W/m$^2$ [26]. Considering the low light emission of c-PI in the atmospheric window, the considerable enhancement of the radiative cooling effects of our metamaterials with a thickness of 50 μm shows the efficacy of the PMMA-SiO$_2$ microstructures.

## Metamaterial-integrated light-emitting diodes and displays

To determine the potency of our metamaterial-integrated optoelectronic devices, the thermal and optical characteristics of light-emitting diodes (LEDs) on which our transparent metamaterials (24 wt% PMMA-SiO$_2$ microstructures) were integrated were experimentally investigated and compared to bare LEDs and c-PI-integrated LEDs as shown in Fig. 4. A commercial blue LED chip covered by a silicone lens was employed for the measurements, and the metamaterials or c-PI were integrated on top of the silicone lens. To estimate the optical characteristics of the displays operating in the outdoor environment, the indoor characterization under illuminated conditions was performed to strictly quantify the light output power of the LED chip in response to the temperature rise under the controlled environmental conditions in terms of solar intensity, ambient temperature, and relative humidity. First, the steady-state temperatures of the LED chips operated with an input power of 1 W for 30 min are monitored in indoor dark conditions, showing that integrating the metamaterial on top of the LED chip suppresses the maximum temperature of bare LED chip by 3.0 °C (Fig. 4a). Despite small differences, the existence of c-PI on the LED chips slightly increases the maximum temperatures of the LED chips because the silicone lens (~0.7 in the wavelengths of 2.5-25 μm[25,59]) on top of the commercial blue LED chip has a slightly higher emissivity than c-PI (~0.6 in the wavelengths of 2.5-25 μm). These results show that radiative cooling by metamaterials can effectively decrease the maximum temperatures of heat-generating electronic devices, even under dark conditions. Previous efforts have focused on the bottom side of LED devices to enhance cooling capacity. This study demonstrates the capability of enhancing the cooling of LED devices by promoting heat dissipation through the front side of LED devices.

The temperature suppression effects become more distinct under 1.5 G solar illumination under indoor conditions (Fig. 4b). Integrating metamaterials on the commercial LED chips considerably decreases the temperature rise of bare LED chips operated with an input power of 1 W for 30 min by 6.7 °C to show excellent radiative cooling effects of our metamaterials. Moreover, equipping metamaterials on top of LED chips enhances their optical characteristics. To investigate the sole optical enhancements by the metamaterials, the light output powers of the samples are monitored as a function of injection currents, while the temperatures of the LED chips are monitored at ~25 °C (Fig. 4c). Measurements were carried out immediately after the LED chip was lightened to minimize the thermalization effects. As a result, the LED chips integrated with the metamaterials (3081 mW) exhibit 1.17 times and 1.15 times higher light output powers than bare LED chips (2629 mW) and c-PI integrated LED chips (2684 mW) at the current of 350 mA. The slightly lower refractive index of c-PI (~1.50) than that of silicone (~1.52) generates anti-reflection effects owing to the graded refractive index, and the high degree of optical haze (~0.64) of the metamaterials promotes light scattering effects to enhance the light output power of the LED chips[60]. The thickness of the adhesive layer plays an important role to minimize thermal resistance of heat conduction for effective radiative cooling[15]. We further investigated the integration method of the metamaterial on the LED chip. Instead of the metamaterial film attachment by using an adhesive layer, the metamaterial layer with the resulting thickness of ~50 μm was directly integrated by using spin coating of the mixture solution on the LED chip covered by the silicone lens. The light output power and the steady-state temperature of the LED chips attached with the metamaterial film by the adhesive layer were compared with those of the

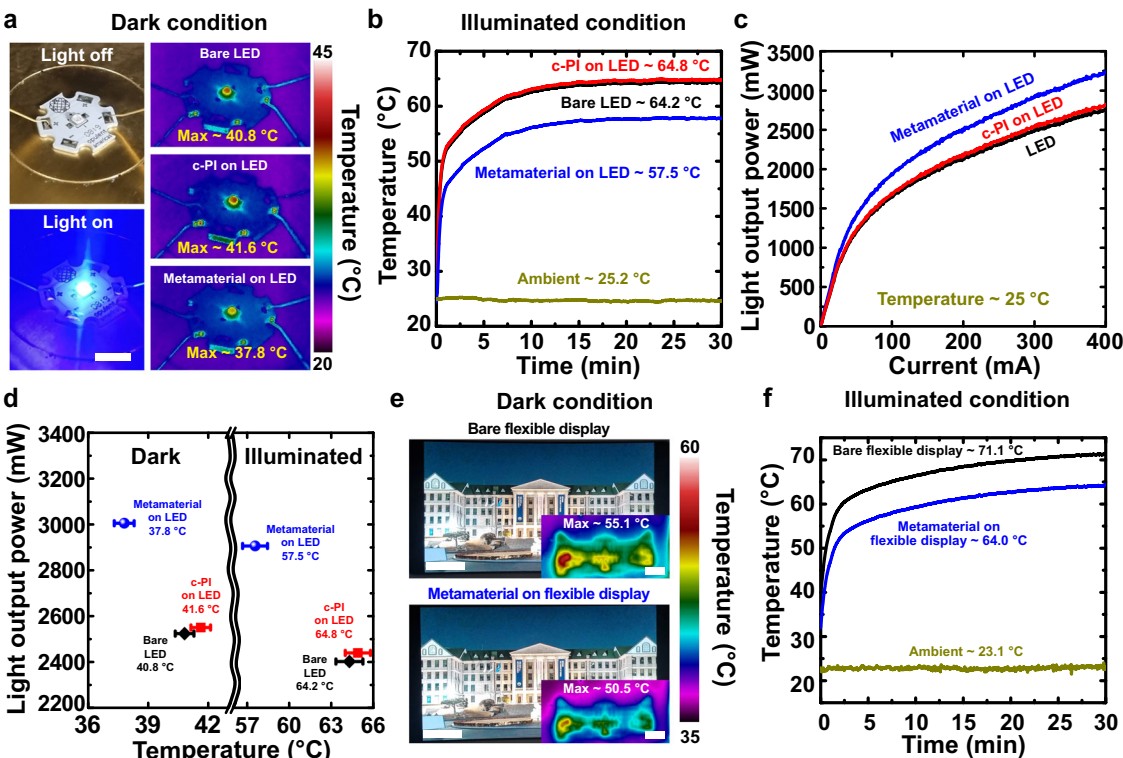

**Fig. 4 | Photographs and performance of commercial LED and flexible display with metamaterial.** The temperatures of bare LED, c-PI on LED, and metamaterial on LED are **a** measured and imaged by an infrared thermal camera under dark condition, and **b** measured under illuminated condition. **c** Light output power as a function of injection current for the LEDs with bare LED, c-PI on LED, and metamatieral on LED, excluding the effect of temperature. **d** Measured light output power versus temperature of the bare LED, c-PI on LED, and metamaterial on LED. Temperatures represent the steady-state temperatures of light-on samples under dark and illuminated conditions. The temperatures of bare flexible display and metamaterial on flexible display are **e** measured and imaged by an infrared thermal camera under dark condition, and **f** measured under illuminated condition. Scale bars: (**a**) 1 cm; (**e**) 2 cm, 2 cm (inset). Error bars in **d** indicate variations in measurements of the samples, displaying the mean and standard distribution ($n = 50$). Source data are provided as a Source Data file.

LED chips integrated with the metamaterial by direct coating method. As a result, they exhibit the similar light output power (within 3%) and steady-state temperatures (within 5% in dak condition and within 1% in illuminated condition) (Supplementary Fig. 13 and Supplementary Table 4), confirming that the adhesive layer does not significantly affect the thermal performance of the metamaterials. The minimized air void between the metamaterial and the device is not only crucial to achieve visible transparency due to the minimized reflection by the light scattering[32], but also important to secure effective suppression of the temperature rise of LED chips, due to the minimized thermal resistance of heat conduction[15]. As a control experiment, the metamaterial film was placed onto the LED chip covered by the silicone lens with air void. Specifically, an adhesive layer was partially applied between the metamaterial film and the silicone lens on the LED chip. The light output power and the steady-state temperature of the LED chips with the metamaterial film (air void) were characterized. As a result, the control sample exhibits the considerably decreased light output power and the increased steady-state temperatures (Supplementary Fig. 13 and Supplementary Table 4). This indicates the importance of the interface between the metamaterial and the device for effective thermal management by the radiative cooling materials. In our study, the LED chip integrated with the silicone lens is employed to characterize the performance of the metamaterials. Specifically, the silicone lens with a radius of 1.5 mm and a height of 1 mm was integrated on the LED chip. When the planar metamaterial film is directly attached on the LED chip without the silicone lens, its planar shape exhibits different light distribution from the hemispherical shape[61], and the absence of the silicone layer leads to the slight sacrifice in the anti-reflection effects generated by the graded refractive index

between silicone (~1.52) and c-PI (~1.50). Nonetheless, the absence of the silicone lens (thermal conductivity of silicone as 0.2–0.4 W/m·K[62]) can considerably decrease thermal resistance of heat conduction, enhancing the radiative cooling effects[15].

Furthermore, decreasing the temperature rise of LED chips using metamaterials greatly enhances optical performance by suppressing the degradation of the light output power in response to the temperature rise[12,14]. Specifically, the light output powers of the samples were compared at a current of 350 mA under indoor dark conditions when the temperatures of the LED chips reached steady-state conditions. To mimic the solar irradiation, the light output powers of the samples are characterized, while the temperature of the sample LED chips is controlled to the observed steady-state temperatures (58 °C–65 °C) in Fig. 4b by controlling the film heater on which the sample LED chips are mounted. While bare LED chips suffer from a decrease in light output power from 2629 mW at ~25 °C through 2529 mW at ~41 °C to 2408 mW at ~64 °C by up to 9.2%, the metamaterial-integrated LED chips exhibit a decrease in light output power (a current of 350 mA) from 3081 mW at ~25 °C through 3007 mW at ~38 °C to 2908 mW at ~58 °C by up to 5.9% (Fig. 4c, Fig. 4d and Supplementary Fig. 14). Considering the temperature effects, mounting the metamaterials on top of the LED chips generates 1.19 times and 1.21 times higher light output powers than bare LED chips under dark and illuminated conditions, respectively. Consequently, the suppression of the temperature rise of the LED chips by metamaterials mitigates the degradation in the optical performance of LED devices owing to thermalization effects. Implementing the microstructured design on the LED surface can enhance light outcoupling efficiency. Previous studies successfully demonstrated that

implementing the microstructures on the LED surfaces increased external quantum efficiency (EQE) from 16% to 22%[63], from 20% to 33%[64], and from 5% to 21%[65]. Meanwhile, increasing EQE from 20% to 33% of the LED accompanies the suppression of the temperature rise by up to 3.8 °C[64]. Although the structured surfaces of LED are effective to reduce the temperature rise, reducing the temperature rise of the LED needs diverse strategies. From that point of view, our proposed metamaterial film can contribute to enhance thermal management of the LED and display devices. Finally, the thermal characteristics of a flexible display panel on which the transparent metamaterial film was mounted were investigated and compared with those of a bare flexible display panel (Fig. 4e, f). While our transparent metamaterials provide clear display images, they successfully decrease the maximum temperature of the flexible display panel by 4.6 °C and 7.1 °C in indoor dark and illuminated conditions, respectively. Further, predicting the temperature response of the simulated display was carried out in comparison to the model real display (Supplementary Fig. 15 and Supplementary Table 5), showing the potential as transparent radiative cooling cover windows for foldable and flexible displays.

## Discussion

The rational design and synthesis of transparent radiative cooling metamaterials have been successfully demonstrated as a cover window for foldable and flexible displays by randomly distributing PMMA-SiO$_2$ microstructures (i.e., PMMA-infiltrated SiO$_2$ aerogel microparticles) into a c-PI film. Although c-PI has been highlighted as a candidate for the cover window of foldable and flexible displays because of its excellent optical, mechanical, and moisture-impermeable properties, it has low light emission (~60.2%) in the atmospheric window, which makes it challenging to achieve a high degree of radiative cooling with a constrained thickness of ~50 μm. Deploying 24 wt% PMMA-SiO$_2$ microstructures to the matrix of c-PI generates four distinct features under the controlled identical film thickness of 50 μm: (1) minimal sacrifice of visible transparency with inclusion of those microstructures owing to refractive index matching effects between PMMA and c-PI to result in high visible transmission of 85.5% in the wavelengths of 400–800 nm, (2) tunable optical haze of cover windows from 0.25 to 0.64 due to the increased light scattering effects caused by the existence of PMMA-SiO$_2$ microstructures, (3) excellent light emission of 94.6% (1.5 times higher than bare c-PI) in the metamaterials which greatly benefits to suppress temperature rise of displays, (4) 2.2 times higher elastic modulus (2.51 GPa) and 1.6 times higher tensile strength (79.8 MPa) of the metamaterials than c-PI, which is favorable to provide the enhanced protection of display from mechanical damages, and (5) 0.6 times lower water vapor transmission rates of the metamaterials than c-PI owing to the increased diffusion of moisture by the microstructures to enhance moisture-impermeable functionality of a cover window. Our 50 μm thick metamaterial films successfully decrease the temperature rise of the simulated devices with heat generation of 100 W/m$^2$ by 6.9 °C in indoor illuminated conditions (AM 1.5 G solar irradiation), which is also confirmed for the simulated devices with heat generation of 400 W/m$^2$ in outdoor condition (a solar intensity of 850 W/m$^2$, ambient temperature of 23–29 °C, and a relative humidity of 30–40%) to demonstrate up to 8.3 °C decrease of the temperature rise of the devices. The metamaterials on top of the LED chips not only enhance the light output power by promoting light scattering but also mitigate the thermalized degradation of optical performance by suppressing the temperature rise of the devices, thereby leading to 1.19 times and 1.21 times higher light output powers of LED chips under dark and illuminated conditions, respectively. Finally, the demonstration of the metamaterial-integrated flexible display panel to suppress the temperature rise by 4.6 °C and 7.1 °C in indoor dark and illuminated conditions, respectively, shows the promise of radiative cooling cover windows for foldable and flexible displays. Our study represents an effort to realize wide applications of radiative cooling technology by proposing effective thermal management of high-performance electronic devices with high-brightness displays for their ubiquitous deployment in indoor and outdoor environments.

## Method

### Synthesis of transparent radiative cooling metamaterials for display cover windows

The PMMA-SiO$_2$ microstructures were synthesized by mixing SiO$_2$ aerogel microparticles (4–10 μm in size, Sigma-Aldrich) and 20 wt% PMMA solution in anisole (950 A2, Kayaku) at a mass ratio of 1:5 using a central mixer (ARE-310, Thinky) at 2000 rpm for 3 min. Then, the mixture was baked at 100 °C for 1 min to remove anisole. Then, c-PI (TPI-100, PNS tech) was then mixed with the prepared PMMA-SiO$_2$ microstructures at the desired mass ratio (0–24 wt% (i.e., 0–16 vol%) PMMA-SiO$_2$ microstructures in c-PI). The prepared mixture was spin-coated onto an aluminum substrate at 700 rpm for 5 min. The samples were cured at 100 °C for 30 min, and 150 °C for 30 min in successive manner. Finally, the cured film was peeled off from the substrate, and the film thickness was approximately 50 μm.

### Measurements of optical, mechanical, and moisture-impermeable properties

UV-VIS-NIR light transmittance and reflectance at wavelengths of 300–2500 nm were measured using a UV-visible spectrometer with an integrating sphere (Lambda 650 S, Perkin Elmer), and IR light transmittance and reflectance in the wavelengths of 2.5–25 μm region were measured using a Fourier transform infrared spectrometer (Nicolet 6700, Thermo Scientific). All samples had a thickness of 50 μm unless otherwise mentioned. The haze in visible light was calculated by dividing the diffuse transmittance by the total transmittance[66]. The diffuse and total transmittances were measured according to the ASTM D1003 method. To quantify the mechanical properties of the metamaterials, tensile tests were performed using a universal testing machine with a 100 N load cell at a speed of 0.1 mm/min. Dogbone-shaped specimens were prepared according to the ASTM D 412 standard. To evaluate the moisture-impermeable properties of the metamaterials, the water vapor transmission rate was measured using a permeation testing system (Permatran-W 3/33 MA, Mocon) at room temperature and at a relative humidity of 100% according to the ASTM F1249 method.

### Indoor characterization of metamaterial-integrated devices

Temperature monitoring was carried out by placing the sample in the middle of an aluminized wood frame (25×25×15 cm$^3$) covered with an optically transparent low-density polyethylene (LDPE) film under AM 1.5 G solar illumination (1000 W/m$^2$) with a solar simulator (Oriel Sol3A, 94023 A, Newport). The cover window films (i.e., c-PI and metamaterial) were attached using an adhesive layer to a plate heater (4 × 4 cm$^2$) to simulate the heat generation of electronic devices. After attaching a thermocouple at the center of the front side of the plate heater, liquid-state c-PI was coated over the plate heater as a thin adhesive layer, after which a c-PI film or metamaterial film with a thickness of ~50 μm was placed and pressed, followed by the curing process. A heating power density of 100 W/m$^2$ was maintained using a power supply (Model 2400, Keithley) connected to the heater. For temperature monitoring of the LED chips and flexible display panels, a commercial blue LED chip (LST1-01F06-RYL1-00, New Energy) and a home-built flexible display panel (TOP060S02K, Wisecoco) were employed, and the cover materials were attached to the top of the LED chips and flexible display panels using the attachment method mentioned above. Specifically, the liquid-state c-PI was spin-coated at 2000 rpm for 3 min as an adhesive layer over the LED chip covered by the silicone lens, after which a c-PI film or a metamaterial film with a thickness of 50 μm was placed and pressed, followed by the curing

process. In this process, the thickness of the adhesive layer was estimated as ~0.5 µm by measuring the thickness difference between the metamaterial film before the attachment and the metamaterial layer after the attachment. In addition, direct coating of the mixture solution on the LED chip covered by the silicone lens was carried out by spin-coating the mixture solution at 600 rpm for 5 min, followed by curing at 100 °C for 30 min and 150 °C for 30 min. The resulting thickness of the cured metamaterial on the LED chip was ~50 µm. Thermocouples (type-T) were attached to the front sides of the heater, LED chip, and display panel. Humidity was quantified using a hygrometer (J-303; Blutec). The temperature was visualized using a thermal imaging camera (T620, FLIR). The light output power characteristics were investigated in terms of the injection currents using an LED tester (ELT-1000, Ecopia).

### Outdoor characterization of metamaterial-integrated devices
The temperature behavior was quantified in Seoul, South Korea, by positioning the samples in the middle of an aluminized wood frame covered with an LDPE film. The frame was tilted 30° southward to allow the normal incidence of the sun to quantify radiative cooling at maximized solar illumination. The samples were prepared by attaching a cover-window film (i.e., c-PI or metamaterial) to a heater with an area of $4 \times 4$ cm$^2$ using the attachment method mentioned above. A heating power density of 400 W/m$^2$ was maintained using a power supply (Model 2400, Keithley) connected to the heater. The temperature at the front surface of the heater (i.e., the bottom surface of the cover-window film) was measured using type-T thermocouples. The solar irradiance was monitored using a pyranometer (SR15, Jinyoung Tech).

### Estimation of net cooling power and temperature response
The net cooling power ($P_{net}$) can be expressed as follows[19,25,29]:

$$P_{net}(T) = P_{rad}(T) - P_{surr}(T_{amb}) - P_{sun} + P_{non-rad}(T, T_{amb}) - P_{gen} \quad (1)$$

where $P_{rad}$ is the radiative cooling power, $P_{surr}$ is the re-emission power from the surrounding, $P_{sun}$ is the thermal power from solar irradiation, $P_{non-rad}$ is the non-radiative cooling power, $P_{gen}$ is the thermal power from device heat generation, $T$ is cooler surface temperature, and $T_{amb}$ is ambient temperature. Specifically, the radiative cooling power ($P_{rad}$) is expressed as follows:

$$P_{rad}(T) = 2\pi \int_0^{\pi/2} \int_0^\infty I_B(T, \lambda) \varepsilon_r(\lambda, \theta) \sin(\theta) \cos(\theta) d\lambda d\theta \quad (2)$$

where $I_B$ represents the spectral radiance of a black body within the wavelength range of 2.5–25 µm at temperature T, while $\varepsilon_r$ denotes the metamaterial emissivity, and the angle $\theta$ is the zenith angle. Metamaterials were assumed to normally face the sun. The re-emission power from the surrounding ($P_{surr}$) is expressed as follows:

$$P_{surr}(T_{amb}) = 2\pi \int_0^{\pi/2} \int_0^\infty I_B(T_{amb}, \lambda) \varepsilon_r(\lambda, \theta) \varepsilon_{surr}(\lambda, \theta) \sin(\theta) \cos(\theta) d\lambda d\theta \quad (3)$$

$\varepsilon_r$ indicates the emissivity of metamaterial. Assuming $T_{amb}$ as 25 °C, for the outdoor environment, $\varepsilon_{surr}$ (the surrounding emissivity) is ~0.2 in the atmospheric window (wavelengths of 8–13 µm) and ~1.0 in other infrared region, and for the indoor environment, $\varepsilon_{surr}$ is 1.0 in all the infrared region (2.5–25 µm) to consider the ceiling and walls as an enclosure. The light absorption of the metamaterial between wavelengths of 300 nm to 2.5 µm was ~3% under solar irradiance, resulting in $P_{sun}$ for the outdoor and indoor environments as ~21 W/m$^2$ and ~23 W/m$^2$, respectively. The non-radiative cooling power ($P_{non-rad}$) is

expressed as follows:

$$P_{non-rad} = h_c(T - T_{amb}) \quad (4)$$

where $h_c$ represents the combined non-radiative heat transfer coefficient. The thermal power from device heat generation ($P_{gen}$) is 0 W/m$^2$ to estimate the net cooling power of transparent radiative cooling metamaterial film alone.

Further, the temperature response of the simulated display is estimated in comparison to the model real display by using the following equation.

$$mc\frac{dT}{dt} = -P_{net}(T) = -\left( P_{rad}(T) - P_{surr}(T_{amb}) - P_{sun} + P_{non-rad}(T, T_{amb}) - P_{gen} \right) \quad (5)$$

where $m$ and $c$ is mass per unit area and heat capacity of target object, respectively, and $t$ is time. It should be noted that $P_{gen}$ is 100 W/m$^2$ in estimating the temperature response of the simulated display and model real display. It should be noted that while calculating the net cooling power is designated to the metamaterial film itself, setting the equation to express the temperature response of the display integrated with the film requires the minus sign to the net cooling power term. It should be also noted that to examine the transient response of the experiments, the estimated temperature response was compared with the experimental results in the indoor environment. The detailed parameter values used in the simulation are listed in Supplementary Table 5.

## Data availability
Source data are provided with this paper.

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

## Acknowledgements

This project was supported by the Basic Science Research Program (NRF-2021R1A2C1011418) and Nano-Material Technology Development Program (NRF-2022M3H4A1A02046445) of the National Research Foundation of Korea (NRF), funded by the Ministry of Science and ICT of Korea.

## Author contributions

D.R.K. and K.W.L. conceived the research and designed the experiments. D.R.K., K.W.L., J.Y., and M.K.K. designed and conducted the experiments and data analysis. D.R.K, K.W.L., and J.Y. characterized the mechanical and optical properties of the materials. All authors analyzed the data and commented on the paper.

## Competing interests

The authors declare no competing interests.
