## [Peer Review File · Nature Communications]

Transparent radiative cooling cover window for flexible and foldable electronic displaysREVIEWER COMMENTS

Reviewer #1 (Remarks to the Author):

In this manuscript, Lee et al proposed a foldable, radiative cooling cover window for outdoor displays using a transparent metamaterial. The metamaterial is synthesized by embedding PMMA-SiO₂ microstructures in c-PI, resulting in minimal sacrifice of visible transparency, tunable optical haze, and excellent light emission in the atmospheric window. Furthermore, this metamaterial also exhibits better mechanical properties and moisture-impermeable properties. Although the proposed technology holds great potential, the technology already exists and detailed comparison and results should be further provided to demonstrate the performance advantages of this solution.

1. The author utilizes clear polyimide as the chosen material. However, the conventional polyimide commonly employed is yellow and opaque. Is it possible to substitute it with PET, PVDF, or other materials?
2. It would be better to provide an SEM or TEM image of the PMMA-SiO₂ microstructures instead of just an illustration diagram in Figure 1a.
3. On page 4, the statement "...increased content of emissive materials (e.g., silica microstructures) in the polymer matrix...". It is not appropriate to classify the silica microstructures as emissive materials. Instead, it would be more suitable to provide the emissivity values of both silica and polymer, and add the references.
4. On page 5, the statement "SiO₂ aerogel microparticles sized 4-10 μm , where SiO₂ nanospheres with a diameter of ~ 20 nm forms a chain-like network, are employed in metamaterials" lacks a related reference, making it confusing. It would be helpful to provide a reference to support the choice of SiO₂ aerogel microparticles with sizes between 4-10 μm . Additionally, it would be beneficial to explain the effect of SiO₂ aerogel microparticle size on emissivity.
5. On page 12, the statement "...While bare LED chips suffered from a decrease in light output power from 2,629 mW at ~ 25 $^{\circ}\text{C}$ through 2,529 mW at ~ 41 $^{\circ}\text{C}$ to 2,408 mW at ~ 65 $^{\circ}\text{C}$ by up to 9.2%, the metamaterial-integrated LED chips exhibited a decrease in light output power (a current of 350 mA) from 3,081 mW at ~ 25 $^{\circ}\text{C}$ through 3,007 mW at ~ 38 $^{\circ}\text{C}$ to 2,908 mW at ~ 58 $^{\circ}\text{C}$ by up to 5.9% (Figure 4d and Supplementary Figure 7)...". The results of the bare LED and metamaterial-integrated LED at ~ 25 $^{\circ}\text{C}$ are not provided in Figures 4d and S7.
6. Can it be experimentally proven that the PMMA-SiO₂ microstructure has better radiative cooling ability than the air-SiO₂ microstructures?
7. The temperature increase during LED operation is primarily caused by low EQE, leading to an increase in LED temperature. Implementing a micro- or nanostructured design on the LED surface can enhance light outcoupling efficiency, thereby reducing the temperature rise during LED operation. This approach is more fundamental and efficient in solving this problem. Therefore, is it necessary to cover the proposed metamaterial film on the chips as described in this paper?
8. In Figure 4, the integration of metamaterials or c-PI on top of the LED chip covered by the silicone lens. Does the fit between the film and the device affect the temperature results?
9. Can these mixture solutions of PMMA-SiO₂ and c-PI be coated directly on the chip to form a metamaterial layer instead of being made into a film?
10. As the metamaterial film is on the silicone lens, what is the effect if it is directly on the LED chip? Will the performance be better?
11. It is suggested to supplement a table showing the details of the advantages of such metamaterial compared to those reported in the literature.
12. In Figure 3a, the unit format of $^{\circ}\text{C}$ is incorrect.
13. On Page 11, a blank is missing in the sentence "...because the silicone lens (~ 0.7 in the wavelengths of 2.5-25 μm ...".

Reviewer #2 (Remarks to the Author):

Lee et al. reported a transparent radiative cooling metamaterials as a cover window of foldable and flexible displays by synthesis of embedding optically-modulating microstructures in polyimide. In my opinion, this research is meaningful because this material has certain practical applications. However, the characteristics of high solar transmittance and high thermal infrared emissivity are not rare, and it is not hard to achieve this performance. Therefore, I think this paper does not meet the high standard of novelty of Nature Communications. I would give comments listed below that I hope would be helpful for the revision.

Comments:

1. The Introduction chapter should explain this study's novelty. Although the authors said that most radiative cooling materials are highly solar-reflective, that is, opaque, many transparent radiative cooling materials have also been reported. What are the advantages of the authors' study compared to transparent cooling materials? What challenges does it solve?
2. In fact, many ordinary polymer films have the characteristics of high sunlight transmission and high thermal infrared emissivity, and do not even need a special structural design, and the material reported by the author although the sunlight transmittance is still ok, but the thermal infrared emissivity is really not high, and the emissivity begins to decline significantly after 10 microns. I think the spectral properties of this material are really mediocre, and the performance may be better if the author can design an outstanding selective emissivity.
3. The performance shown in Figure 3 and Figure 4 is only compared with c-PI (no PMMA-SiO₂), which is not comprehensive and meaningless, just only showing the improvement of performance by increasing PMMA-SiO₂ wt%. Readers would like to see the advantages of this material in the current field. Authors can increase the performance comparison with the current commercial films and the transparent cooling films reported in the past literature.
4. I suspect that even an unmodified polymer film, such as PDMS, would have a higher solar transmittance and thermal infrared emissivity. Many polymers themselves contain rich chemical bonds that can stretch and vibrate in the transparency atmospheric window.

Reviewer #3 (Remarks to the Author):

To develop a foldable and flexible cover window with optimized radiative cooling properties, the authors dispersed PMMA-infiltrated SiO₂ aerogel microparticles into clear polyimide (c-PI). Through rational design, the cover window showed improved thermal emission in the atmospheric window with secured optical transparency as well as enhanced mechanical and moisture-impermeable properties for potential applications. The radiative cooling characteristics of the window cover in indoor and outdoor conditions were experimentally investigated. Furthermore, the authors demonstrated metamaterial-integrated light-emitting diodes and displays. Overall, this work is well-organized and the results seem promising. However, several issues should be addressed to further improve this manuscript.

1. Please explain what sample (SiO₂ wt%) was used to get the images in Fig. 1b-d and the data in Fig. 1e.
2. In the 1st paragraph of page 7, the authors claimed that an increased optical haze enables "broad applications of displays from clear visualization to anti-glare effects".
 - a) The optical haze of 0.64 might not be the optimal value for screens. The image in Fig. 1c seems quite blurry. Please specifically compare the optical haze of this work with existing technologies. It encouraged to list the optical haze of commercialized products.
 - b) The sample with highest PMMA-SiO₂ wt% (24 wt%) has the best radiative cooling performance and the highest optical haze (0.64). Though the optical haze of 0.64 might be too high for applications, the authors used this sample (24 wt%) in almost all following experiments (Fig. 2e, Fig. 3 and Fig. 4). What would the radiative cooling performance of other samples (with lower PMMA-SiO₂ wt%) in those experiments?
3. In the 2nd paragraph of page 7, though adding PMMA-infiltrated SiO₂ aerogel microparticles could help boost the elastic moduli and tensile strengths of c-PI. However, it is still unclear whether

this work has fulfilled all the mechanical requirements for practical applications. What are the desired elastic moduli and tensile strengths in this research field? A detailed comparison between this work and commercialized products is necessary.

4. In Fig. 2e, the authors demonstrated foldable properties of their work. However, a complete lifecycle test is critical in order to claim "excellent foldable properties upon repeated folding and releasing processes".

5. In Fig. 4a and 4b, the authors showed some indoor experimental results.

a) Due to the radiative heat transfer between the radiative cooler and ceiling/walls, indoor radiative cooling power is weaker compared to outdoor counterparts. It would be helpful if the net cooling power of this indoor experiment could be calculated, preferable with numerical modeling.

b) The authors used "simulated display" in the experiment. However, several key parameters (heat capacity, mass, etc.) of this simulated display might be different from those of a real display. What would be the temperature drop if a real display is used in the experiment?

c) Please clarify the reason for using AM 1.5G solar irradiation in the indoor experiments. The solar irradiation in a room is normally weaker than outdoor environments.

6. Please specify the PMMA-SiO₂ wt% of the sample that was tested in the section of "Metamaterial-integrated light-emitting diodes and displays".

7. In the "Measurements of optical, mechanical, and moisture-impermeable properties" of the Method section, the authors used a UV-visible spectrometer and a FTIR spectrometer to characterize the cover window. It is encouraged to specify whether integrating spheres were used.

8. In the "Indoor characterization of metamaterial-integrated devices" of the Method section, the authors used liquid-state c-PI as a thin adhesive layer. Since thickness is a key factor in this work, please provide the thickness of the c-PI adhesive layer. What is the thermal influence of this adhesive layer?

Several minor comments include:

1. The arrow in Fig. 2a is pointing to the wrong direction.
2. In Fig. 3b, there is a typo in the title of right Y-axis.

We thank all the reviewer #1, #2 and #3 for the careful review. Our response to the point raised in the report is as follows.

Reviewer #1 (Remarks to the Author):

Comment #1

In this manuscript, Lee et al proposed a foldable, radiative cooling cover window for outdoor displays using a transparent metamaterial. The metamaterial is synthesized by embedding PMMA-SiO₂ microstructures in c-PI, resulting in minimal sacrifice of visible transparency, tunable optical haze, and excellent light emission in the atmospheric window. Furthermore, this metamaterial also exhibits better mechanical properties and moisture-impermeable properties. Although the proposed technology holds great potential, the technology already exists and detailed comparison and results should be further provided to demonstrate the performance advantages of this solution.

Response and change

Thank you for the positive comments. For the successful implementation of the radiative cooling functionality to the cover windows of foldable and flexible displays, the radiative cooling materials need to meet the requirements of high optical transparency, suitable mechanical property (foldability), and excellent moisture impermeability. From that point of view, we compared our transparent metamaterials with the previous “transparent” radiative cooling materials in terms of their key features and radiative cooling performance in Table A [#1-1~#1-20]. Above all, the cover window materials of foldable and flexible displays should have high elastic modulus, enabling the excellent elastic recovery and rebound resilience [#1-21]. While SiO₂-based transparent radiative cooling materials are stiff and brittle for the cover windows of foldable displays, PDMS-based transparent radiative cooling materials do not meet the foldability requirement for the displays due to the considerably low elastic modulus. As our response to Comment #15, the repeated bending tests greatly generated the wrinkles on PDMS, showing that PDMS did not meet the requirement of the cover windows of foldable and flexible displays. In addition, TPX, silk fibroin and cellulose-based transparent radiative cooling materials have poor moisture impermeability. To the best of our knowledge, this study is first demonstration of excellent radiative cooling performance of the metamaterials which can address the requirements of the cover windows of foldable and flexible displays. **We have added the discussions to our manuscript for the better clarity.**

Reference

[#1-1~#1-20] Science 355 1062-1066 (2017); PNAS 112 40 12282-12287 (2015); Small 15 1905290 (2019); Renewable Energy 191 662-668 (2022); Light Sci. Appl. 11 122 (2022); Adv. Funct. Mater. 33 2301924 (2023); Cell Rep. Phys. Sci. 100231 (2020); Adv. Funct. Mater. 32 2105882 (2022); Solar energy 225 245-251 (2021); Cellulose 28 9383-9393 (2021); Membranes 10 9 (2020); Materials 10 7 821 (2017); J. Micro electro mechanical Sys. 19 2 229-238 (2010); Thin Solid Films 720 138524 (2021); Adv. Funct. Mater. 28 1705291 (2018); *Materials* 12, 3340 (2019); J. Appl. Polym. Sci., 131 41050 (2014); IEEE International Conference on MEMS 397-400 (2015); Industrial Crops and Products 138 111478 (2019); Carbohydrate Polymers 111 442-448 (2014).
[#1-21] Xie J. *et al.* Highly Foldable, Super-Sensitive, and Transparent Nanocellulose/Ceramic/Polymer Cover Windows for Flexible OLED Displays. *ACS Applied Materials & Interfaces* **14**, 16658-16668 (2022).

Table A. Comparison of the recent studies on the transparent radiative cooling materials.

Material	Transmission in visible wavelengths	Emissivity in atmospheric window	Solar intensity / Relative humidity	Temperature drop (control sample)	Elastic modulus	WVTR	Reference
SiO ₂ microspheres in TPX	-	>0.93	~900 W/m ² /-	-	~360 MPa* (TPX)	~775 g/m ² ·day [†] (TPX)	[#1-1]
SiO ₂ photonic crystal arrays	-	>0.90	~1000 W/m ² /-	13 °C (Si wafer)	130~188 GPa [‡] (SiO ₂)	~2·10 ⁻³ g/m ² ·day [§] (SiO ₂)	[#1-2]
SiO ₂ microspheres on glass	~84%	>0.98	800 W/m ² / ~37%	14 °C (Si wafer)	130~188 GPa [‡] (SiO ₂)	~2·10 ⁻³ g/m ² ·day [§] (SiO ₂)	[#1-3]
Grating SiO ₂ structure	~90%	~0.90	830~990 W/m ² /30~40%	3.6 °C (Si wafer)	130~188 GPa [‡] (SiO ₂)	~2·10 ⁻³ g/m ² ·day [§] (SiO ₂)	[#1-4]
SiO ₂ /TiO ₂ multi-layer on ITO-coated PET film	~87%	~0.88	~700 W/m ² /-	9.6 °C (Roof)	130~188 GPa [‡] (SiO ₂)	~2·10 ⁻³ g/m ² ·day [§] (SiO ₂)	[#1-5]
Silk fibroin	~91%	~0.88	~900 W/m ² /-	5.1 °C (Solar panel)	~1 GPa** (Silk fibroin)	2,000 g/m ² ·day ^{††} (Silk fibroin)	[#1-6]
PDMS on ITO-coated window	~94%	~0.90	~630 W/m ² / ~13%	7 °C (Bare window)	~2.61 MPa ^{‡‡} (PDMS)	708 g/m ² ·day ^{§§} (PDMS)	[#1-7]
n-hexadecane-infiltrated SiO ₂ aerogel microparticles in PDMS	~91%	~0.98	~920 W/m ² / 30~45%	7.7 °C (Solar cell)	~2.61 MPa ^{‡‡} (PDMS)	708 g/m ² ·day ^{§§} (PDMS)	[#1-8]
Microstructured PDMS	>90%	Close to 1.0	-	4 °C (Solar cell)	~2.61 MPa ^{‡‡} (PDMS)	708 g/m ² ·day ^{§§} (PDMS)	[#1-9]
Nanofibrillated cellulose film	>90%	>0.85	~800 W/m ² / ~27%	5 °C (Si wafer)	46 GPa*** (Cellulose)	1,315 g/m ² ·day ^{†††} (Cellulose)	[#1-10]
PMMA-SiO₂ microstructures in c-PI	~85%	~0.95	~900 W/m² / 30~40%	8.3 °C (c-PI on simulated display (400W/m²))	2.51 GPa	66 g/m²·day	Our work

* Elastic modulus of TPX: [#1-11], † WVTR of TPX: [#1-12], ‡ Elastic modulus of SiO₂: [#1-13], § WVTR of SiO₂: [#1-14], ** Elastic modulus of silk fibroin: [#1-15], †† WVTR of silk fibroin: [#1-16], ‡‡ Elastic modulus of PDMS: [#1-17], §§ WVTR of PDMS: [#1-18], *** Elastic modulus of cellulose: [#1-19], ††† WVTR of cellulose: [#1-20]

Our modification to the manuscript

To clarify this, we have added Table A to the manuscript as **Supplementary Table 1**.

Also, we have revised the manuscript **on line 10 of page 5**, **“We compare our transparent metamaterials with the previously demonstrated transparent radiative cooling materials in terms of their key features and radiative cooling performance in Supplementary Table 1. The cover window materials of foldable and flexible displays should have high elastic modulus, enabling the excellent elastic recovery and rebound resilience³. While transparent radiative cooling materials based on SiO₂ are stiff and brittle for the cover windows of foldable displays, PDMS-based transparent radiative cooling materials do not meet the folding requirement due to the considerably low elastic modulus. TPX, Silk fibroin and cellulose-based transparent radiative cooling materials have poor moisture impermeability. To the best of our knowledge, this study is the first demonstration of excellent radiative cooling performance of the metamaterials which can address the requirements of the cover windows of foldable and flexible displays.”**

Comment #2

1. The author utilizes clear polyimide as the chosen material. However, the conventional polyimide commonly employed is yellow and opaque. Is it possible to substitute it with PET, PVDF, or other materials?

Response and change

While conventional polyimides are opaque due to their aromatic functional groups [#2-1], clear polyimides (c-PIs), in which the aromatic functional groups are replaced with trifluoromethyl group, are widely used as cover window materials in display applications [#2-2~#2-4]. Our strategy to synthesize the metamaterials with clear polyimide (c-PI) can be applied to PET or PVDF by replacing the optically-modulating material (PMMA) with others to consider refractive index matching, low viscosity, low surface energy, and similar solubility.

Nonetheless, replacing c-PI with PET or PVDF needs further consideration in terms of the requirements of cover window materials for display applications as our response to Comment #1. Specifically, the cover window materials for foldable display applications mainly require high optical transparency, suitable mechanical property (foldability), and excellent moisture impermeability. In addition, touch sensitivity and thermal stability (manufacturing process) should be considered for diverse display applications. From that point of view, we summarized the characteristics of various pristine materials in Table B below [#2-5~#2-36]. PET has weak thermal stability, which requires additional process, including heat resistant coating, during the display manufacture process. PVDF has both weak thermal stability and poor touch sensitivity. **We have added the discussions to our manuscript for the better clarity.**

Table B. Comparison of pristine materials in terms of optical transparency, mechanical property (foldability at a bending radius of 1 mm), moisture impermeability, thermal stability, and touch sensitivity for the cover windows of foldable and flexible displays.

Pristine material	Optical transparency (Visible wavelengths)	Mechanical property (Foldability @ bending radius of 1 mm)	Moisture impermeability (g/m ² day)	Touch sensitivity (Dielectric constant @ 1×10 ⁶ Hz)	Thermal stability
Ultra-thin glass	Good (>90%) [†]	Weak* (Elastic modulus ~70 GPa, Tensile strength ~3 GPa) [‡]	Good (~0.002) [§]	Good (~3.9) ^{**}	Good (T _g = ~552 °C) ^{††}
Clear polyimide	Good (~90%) [†]	Good (Elastic modulus 1~2 GPa, Tensile strength 60-100 MPa) [‡]	Good (~165) [§]	Good (3-4) ^{**}	Good (T _g > 400 °C) ^{††}
PDMS	Good (~95%) [†]	Weak (Elastic modulus 1.3~3.0 MPa, Tensile strength 3.5~7.7 MPa) [‡]	Weak (~708) [§]	Fair (~2.4) ^{**}	Weak (T _g = -127~-121 °C, Boiling point 155~200 °C) ^{††}
PET	Good (>90%) [†]	Good (Elastic modulus ~3 GPa, Tensile strength ~80 MPa) [‡]	Good (~18) [§]	Fair (~2.2) ^{**}	Weak (T _g = 67~81 °C) ^{††}
PVDF	Good (80-90%) [†]	Good (Elastic modulus 2.5-3.5 GPa, Tensile strength 40-60 MPa) [‡]	Good (~29) [§]	Weak (~0) ^{**}	Weak (T _g = -40 °C, Melting point 171~180 °C) ^{††}
TPX	Good (>90%) [†]	Weak (Elastic modulus ~360 MPa, Tensile strength ~20 MPa) [‡]	Weak (~775) [§]	Fair (~2.0) ^{**}	Weak (T _g = 20 ~ 30 °C) ^{††}
Cellulose	Good (85-90%) [†]	Good (Elastic modulus ~46 GPa, Tensile strength 2-3 GPa) [‡]	Weak (~1,315) [§]	Good (~3.0) ^{**}	Weak (T _g = ~107 °C) ^{††}

* Foldability of ultra-thin glass (a thickness of ~ 85 μm) was reported at a bending radius of 3 mm [#2-6].

† Optical transparency: Ultra-thin glass[#2-5], Clear polyimide[#2-10~11], PDMS[#2-13], PET[#2-18], PVDF[#2-22], TPX[#2-27], Cellulose[#2-32]

‡ Elastic modulus, Tensile strength: Ultra-thin glass[#2-6], Clear polyimide[#2-10~11], PDMS[#2-14], PET[#2-19], PVDF[#2-23], TPX[#2-28], Cellulose[#2-33]

§ Water vapor transmission rate: Ultra-thin glass[#2-7], Clear polyimide[#2-12], PDMS[#2-15], PET[#2-20], PVDF[#2-24], TPX[#2-29], Cellulose[#2-34]

** Dielectric constant: Ultra-thin glass[#2-8], Clear polyimide[#2-10], PDMS[#2-13], PET[#2-21], PVDF[#2-25], TPX[#2-30], Cellulose[#2-35]

†† Thermal stability: Ultra-thin glass[#2-9], Clear polyimide[#2-11], PDMS[#2-17], PET[#2-19], PVDF[#2-26], TPX[#2-31], Cellulose[#2-36]

Our modification to the manuscript

To clarify this, we have added Table B to the manuscript as **Supplementary Table 3**.

Also, we have revised the manuscript **on line 18 of page 10**, “**Our strategy to synthesize the metamaterials with c-PI can be applied to other materials (e.g., PET, PVDF, etc.) by replacing the optically-modulating material (PMMA) with others to consider refractive index matching, low viscosity, low surface energy, and similar solubility. For the successful implementation of our strategy to the cover windows of diverse foldable and flexible displays, considering touch sensitivity and thermal stability is also desirable in addition to optical transparency, mechanical property (foldability), and moisture impermeability. From that point of view, we summarize the characteristics of various pristine materials in Supplementary Table 3.**”

Reference

[#2-1] Bai W. *et al.* Solubility, thermal and photoluminescence properties of triphenyl imidazole-containing polyimides. *RSC Advances* **11**, 23802-23814 (2021).

[#2-2] Xie J. *et al.* Highly Foldable, Super-Sensitive, and Transparent Nanocellulose/Ceramic/Polymer Cover Windows for Flexible OLED Displays. *ACS Applied Materials & Interfaces* **14**, 16658-16668 (2022).

[#2-3] Ahn C. *et al.* Highly Transparent, Colorless Optical Film with Outstanding Mechanical Strength and Folding Reliability Using Mismatched Charge-Transfer Complex Intensification. *Advanced Functional Materials* **32**, 2111040 (2022).

[#2-4] Lee Y. *et al.* Transparent and flexible hybrid cover window film: Hard coating/substrate all-in-one composite film for reliable foldable display. *Composites Part B: Engineering* **247**, 110336 (2022).

[#2-5~#2-36] *Progress in Organic Coatings* 187 108162 (2024); *Solar Energy Materials and Solar Cells* 166 254-261 (2017); *J. Micromech. Microeng.* 23 075001 (2013); *J. Materials Research* 28 11 1490-1497 (2013); *J. Non-Crystalline Solids* 324 277-288 (2003); *ACS Appl. Mater. Interfaces* 14 14 16658-16668 (2022); *Advanced Materials Interfaces* 7 2000928 (2020); *Adv. Eng. Mater.* 18 582-590 (2016); *Adv. Funct. Mater.* 32 2105882 (2022); *J. Micromechanics and Microengineering* 24 035017 (2014); *IEEE International Conference on MEMS* 397-400 (2015); *Adv. Funct. Mater.* 30 4 (2020); *Polymers* 13 7 1080 (2021); *Applied Surface Science* 254 3504-3508 (2008); *Composites Part B: Eng.* 37 399-407 (2006); *Surface and Coatings Technology* 206 318-324 (2011); *J. Appl. Polym. Sci.* 132 42508 (2015); *New Biotechnology* 69, 28-35 (2022); *Ferroelectrics* 226 169-181 (1999); *Polymer* 54 1679-1690 (2013); *Sci. Adv.* 3 e1602902 (2017); *Polymers* 14 4793 (2022); *Science* 355 1062-1066 (2017); *Membranes* 10 9 (2020); *Materials* 10 7 821 (2017); *J. Mater. Sci.: Mater. Electron.* 26 9396–9401 (2015); *Phys. Status Solidi (c)* 6 2420-2422 (2009); *ACS Sustainable Chem. Eng.* 10 32 10560–10569 (2022); *Industrial Crops and Products* 138 111478 (2019); *Carbohydrate Polymers* 111 442-448 (2014); *Adv. Funct. Mater.* 30 1904536 (2020); *Cellulose* 15 445–451 (2008).

Comment #3

2. It would be better to provide an SEM or TEM image of the PMMA-SiO₂ microstructures instead of just an illustration diagram in Figure 1a.

Response and change

As the reviewer suggested, we have revised Figure 1a by adding a TEM image of the PMMA-SiO₂ microstructures.

Our modification to the manuscript

To clarify this, we have revised **Fig. 1a** of our manuscript as below.

Fig. 1 Transparent radiative cooling materials for foldable cover window. a Schematical illustration of working principle and applications. PMMA-SiO₂ microstructures, an emissive material, provide radiative cooling functionality with minimal sacrifice of optical transparency. The upper schematic of PMMA-SiO₂ microstructures represents the transmission electron microscopy (TEM) image of PMMA-SiO₂ microstructures.

Comment #4

3. On page 4, the statement "...increased content of emissive materials (e.g., silica microstructures) in the polymer matrix...". It is not appropriate to classify the silica microstructures as emissive materials. Instead, it would be more suitable to provide the emissivity values of both silica and polymer, and add the references.

Response and change

Thank you for the reviewer's comment. As the reviewer suggested, we have added the emissivity values of both silica [#4-1] and polymer [#4-2] with the references to the sentence in our manuscript.

Reference

[#4-1] Wang X. *et al.* Scalable Flexible Hybrid Membranes with Photonic Structures for Daytime Radiative Cooling. *Advanced Functional Materials* **30**, 1907562 (2020).

[#4-2] Srinivasan A, Czapla B, Mayo J, Narayanaswamy A. Infrared dielectric function of polydimethylsiloxane and selective emission behavior. *Applied Physics Letters* **109**, (2016).

Our modification to the manuscript

To clarify this, we have revised the manuscript on line 2 of page 4, "Although the increased content of silica (emissivity of $\sim 0.94^{23}$) microstructures in the polymer matrix (e.g., PDMS, emissivity of $\sim 0.92^{30}$) enhances light emission in the atmospheric window, it inevitably sacrifices visible clarity."

Comment #5

4. On page 5, the statement "SiO₂ aerogel microparticles sized 4-10 μm , where SiO₂ nanospheres with a diameter of ~ 20 nm forms a chain-like network, are employed in metamaterials" lacks a related reference, making it confusing. It would be helpful to provide a reference to support the choice of SiO₂ aerogel microparticles sizes between 4-10 μm . Additionally, it would be beneficial to explain the effect of SiO₂ aerogel microparticle size on emissivity.

Response and change

Thank you for the reviewer's comment. SiO₂ aerogel microparticles are formed as a chain-like network [#5-1] with the SiO₂ nanospheres with a diameter of ~ 20 nm as shown in a schematic (Figure A(a)) and a TEM image (Figure A(b)). While SiO₂ microparticles induce Mie scattering to enhance light emission in the atmospheric window [#5-2, #5-3], SiO₂ microparticles have different extinction coefficients in terms of their sizes [#5-3, #5-4]. Therefore, SiO₂ aerogel microparticles sized 4-10 μm contribute to increase light emission at broadband wavelengths. We have added the detailed information with the related references to our manuscript for the better clarity.

Reference

[#5-1] Salimian S, Zadhoush A, Naeimirad M, Kotek R, Ramakrishna S. A review on aerogel:

3D nanoporous structured fillers in polymer-based nanocomposites. *Polymer Composites* **39**, 3383-3408 (2018).

[#5-2] Mandal J. *et al.* Hierarchically porous polymer coatings for highly efficient passive daytime radiative cooling. *Science* **362**, 315-319 (2018).

[#5-3] Wang X. *et al.* Scalable Flexible Hybrid Membranes with Photonic Structures for Daytime Radiative Cooling. *Advanced Functional Materials* **30**, 1907562 (2020).

[#5-4] Zhai Y. *et al.* Scalable-manufactured randomized glass-polymer hybrid metamaterial for daytime radiative cooling. *Science* **355**, 1062-1066 (2017).

Figure A. **a**, Schematic to illustrate a SiO₂ aerogel microparticle. **b**, Transmission Electron Microscopy (TEM) image to show SiO₂ aerogel microparticles. **c**, Schematic to illustrate a PMMA-SiO₂ microstructure. **d**, TEM image to show PMMA-SiO₂ microstructures.

Our modification to the manuscript

To clarify this, we have added Figure A to the manuscript as **Supplementary Figure 1**.

Also, we have revised the manuscript **on line 26 of page 5**, **"First, the phonon-polariton resonance of SiO₂ at a wavelength of 9.7 μm plays an important role in emitting light in the atmospheric window, while shaping SiO₂ into microstructures enables Mie scattering to**

enhance further the light emission at those wavelength^{23,32}. . . . Since SiO₂ microstructures have different extinction coefficients in terms of their sizes^{23,26}, the broad distribution of SiO₂ microstructures contributes to increased light emission at broadband wavelengths, suppressing the temperature increase^{23,26}. Hence, SiO₂ aerogel microparticles sized 4-10 μm, where SiO₂ nanospheres with a diameter of ~20 nm form a chain-like network³⁴ (Supplementary Figure 1), are employed in metamaterials.”

Comment #6

5. On page 12, the statement “...While bare LED chips suffered from a decrease in light output power from 2,629 mW at ~25 °C through 2,529 mW at ~ 41 °C to 2,408 mW at ~65 °C by up to 9.2%, the metamaterial-integrated LED chips exhibited a decrease in light output power (a current of 350 mA) from 3,081 mW at ~25 °C through 3,007 mW at ~ 38 °C to 2,908 mW at ~58 °C by up to 5.9% (Figure 4d and Supplementary Figure 7)...”. The results of the bare LED and metamaterial-integrated LED at ~ 25 °C are not provided in Figures 4d and S7.

Response and change

Thank you for the reviewer’s comment. Although the results of the bare LED and metamaterial-integrated LED at ~25 °C were provided in Figure 4c, the note was missing in the statement. We have corrected the sentence for the better clarity.

Our modification to the manuscript

To clarify this, we have revised the manuscript on line 9 of page 16, “While bare LED chips suffered from a decrease in light output power from 2,629 mW at ~25 °C through 2,529 mW at ~ 41 °C to 2,408 mW at ~65 °C by up to 9.2%, the metamaterial-integrated LED chips exhibited a decrease in light output power (a current of 350 mA) from 3,081 mW at ~25 °C through 3,007 mW at ~ 38 °C to 2,908 mW at ~58 °C by up to 5.9% (Figure 4c, Figure 4d and Supplementary Figure 7).”

Comment #7

6. Can it be experimentally proven that the PMMA-SiO₂ microstructure has better radiative cooling ability than the air-SiO₂ microstructures?

Response and change

As the reviewer suggested, we experimentally investigated the emissivity and the radiative cooling performance of the air-SiO₂ microstructures in c-PI under the identical composition of SiO₂ microstructures and the identical film thickness of ~50 μm, compared to our metamaterial. The emissivity of the air-SiO₂ microstructures is ~0.90 in the atmospheric window, while the metamaterial exhibits the emissivity of ~0.94. The radiative cooling performance was characterized in indoor conditions by attaching the c-PI film with the air-SiO₂ microstructures on the simulated display with heat generation of 100 W/m². As shown in Figure B, the metamaterial suppresses the temperature rise by 6.9 °C, while the air-SiO₂ microstructure sample exhibits the suppression of the temperature rise by 4.9°C, showing the better radiative cooling performance of the metamaterial. In addition, we investigated the transmission and haze factor in the visible wavelengths of the air-SiO₂ microstructures in c-PI. As shown in Figure C and Figure D, the inclusion of the air-SiO₂ microstructures to c-PI not only considerably sacrifices the transmission in the visible wavelengths, but also greatly increases the haze factor even with their small inclusion due to their light scattering effects. We have added the detailed information with the related references to our manuscript for the better clarity.

Figure B. **a**, Emissivity in the atmospheric window for metamaterial (24 wt% PMMA-SiO₂ microstructures), 24 wt% air-SiO₂ microstructures in c-PI, and bare c-PI. All the samples have an identical thickness of 50 μm. **b**, Radiative cooling performance of metamaterial (24 wt% PMMA-SiO₂ microstructures), 24 wt% air-SiO₂ microstructures in c-PI, and bare c-PI placed on top of the simulated display with heat generation of 100 W/m² in indoor conditions.

Figure C. Optical characteristics of metamaterial (24 wt% PMMA-SiO₂ microstructures), 24 wt% air-SiO₂ microstructures in c-PI, and bare c-PI in visible wavelengths. **a**, Total transmittance and **b**, haze factor in visible wavelengths (400-800 nm). All the samples have 50 μm thickness. Scale bar is 2 cm.

Figure D. Optical characteristics of the air-SiO₂ microstructures in c-PI in terms of air-SiO₂ microstructures contents (0-24 wt%) in visible wavelengths. a, Total transmittance and b, haze factor in visible wavelengths (400-800 nm). All the samples have 50 μm thickness.

Our modification to the manuscript

To clarify this, we have added Figure B, Figure C, and Figure D to the manuscript as Supplementary Figure 9, Supplementary Figure 2, and Supplementary Figure 3, respectively.

Also, we have revised the manuscript on line 1 of page 8, “It should be noted that the inclusion of the air-SiO₂ microstructures to c-PI not only considerably sacrifices the transmission in the visible wavelengths, but also greatly increases the haze factor even with their small inclusion due to their light scattering effects (Supplementary Figure 2 and Supplementary Figure 3).”

Further, we have revised the manuscript on line 22 of page 11, “As a control experiment, the radiative cooling performance of the 24 wt% air-SiO₂ microstructures in c-PI under the identical film thickness was characterized on the simulated display with heat generation of 100 W/m². As shown in Supplementary Figure 9, the air-SiO₂ microstructure sample exhibits the suppression of the temperature rise by 4.9°C, showing the better radiative cooling performance of the metamaterial.”

Comment #8

7. The temperature increase during LED operation is primarily caused by low EQE, leading to an increase in LED temperature. Implementing a micro- or nanostructured design on the LED surface can enhance light outcoupling efficiency, thereby reducing the temperature rise during LED operation. This approach is more fundamental and efficient in solving this problem. Therefore, is it necessary to cover the proposed metamaterial film on the chips as described in this paper?

Response and change

As the reviewer pointed out, implementing the micro- or nanostructured design on the LED surface can enhance light outcoupling efficiency. Previous studies successfully demonstrated that implementing the microstructures on the LED surfaces increased EQE from 16% to 22% [#8-1], from 20% to 33% [#8-2], from 5% to 21% [#8-3]. Meanwhile, increasing the EQE from 20% to 33% of the LED accompanies the suppression of the temperature rise by up to 3.8 °C [#8-2]. Although the structured surfaces of LED are effective to reduce the temperature rise,

reducing the temperature rise of the LED needs diverse strategies because increasing the EQE only cannot fully resolve the thermalization issues of the LED devices. Placing the metamaterial film on the LED chips not only enhances the light output power due to high haze factor, but also suppresses the temperature rise by 3.0 °C and by 6.7 °C under dark and illuminated conditions, respectively, thereby achieving the equivalent effects of further increase of the light output power. From that point of view, our proposed metamaterial film can improve the thermal management of the LED and display devices. **We have added the discussions to our manuscript for the better clarity.**

Reference

- [#8-1] Chu Z. *et al.* Perovskite Light-Emitting Diodes with External Quantum Efficiency Exceeding 22% via Small-Molecule Passivation. *Advanced Materials* **33**, 2007169 (2021).
[#8-2] Cheng, WC. *et al.* AlGaInP Red LEDs with Hollow Hemispherical Polystyrene Arrays. *Sci Rep* **8**, 911 (2018).
[#8-3] Chiba T. *et al.* Anion-exchange red perovskite quantum dots with ammonium iodine salts for highly efficient light-emitting devices. *Nature Photonics* **12**, 681-687 (2018).

Our modification to the manuscript

To clarify this, we have revised the manuscript on line 18 of page 16, **“Implementing the micro-structured design on the LED surface can enhance light outcoupling efficiency. Previous studies successfully demonstrated that implementing the microstructures on the LED surfaces increased external quantum efficiency (EQE) from 16% to 22%⁶³, from 20% to 33%⁶⁴, from 5% to 21%⁶⁵. Meanwhile, increasing EQE from 20% to 33% of the LED accompanies the suppression of the temperature rise by up to 3.8 °C⁶⁴. Although the structured surfaces of LED are effective to reduce the temperature rise, reducing the temperature rise of the LED needs diverse strategies. From that point of view, our proposed metamaterial film can contribute to enhance thermal management of the LED and display devices.”**

Comment #9

8. In Figure 4, the integration of metamaterials or c-PI on top of the LED chip covered by the silicone lens. Does the fit between the film and the device affect the temperature results?
9. Can these mixture solutions of PMMA-SiO₂ and c-PI be coated directly on the chip to form a metamaterial layer instead of being made into a film?
10. As the metamaterial film is on the silicone lens, what is the effect if it is directly on the LED chip? Will the performance be better?

Response and change

For the sample preparation in Figure 4, the liquid-state c-PI was spin-coated at 2,000 rpm for 3 min as an adhesive layer over the LED chip covered by the silicone lens, after which a c-PI film or a metamaterial film with a thickness of 50 μm was placed and pressed, followed by the curing process. In this process, the thickness of the adhesive layer was estimated as ~0.5 μm. Instead of the film attachment, we can also directly coat the mixture solution on the LED chip covered by the silicone lens by spin-coating the mixture solution at 600 rpm for 5 min, followed by curing at 100 °C for 30 min and 150 °C for 30 min. The resulting thickness of the cured metamaterial on the LED chip was ~50 μm. The light output power and the steady-state temperature of the LED chips attached with the metamaterial film by the adhesive layer were compared with those of the LED chips integrated with the metamaterial by direct coating method. As a result, they exhibited the similar light output power (within 3%) and steady-state

temperatures (within 5% in dark condition and within 1% in illuminated condition) (Figure E and Table C below).

The minimized air void between the film and the device is not only crucial to achieve visible transparency due to the minimized reflection by the light scattering [#9-1], but also important to secure effective suppression of the temperature rise of LED chips, due to the minimized thermal resistance of heat conduction [#9-2]. As control experiments, the metamaterial film was placed onto the LED chip covered by the silicone lens with air void. Specifically, an adhesive layer was partially applied between the metamaterial film and the silicone lens on the LED chip. The light output power and the steady-state temperature of the LED chips with the metamaterial film (air void) were characterized. As a result, the control sample exhibited the considerably decreased light output power and the increased steady-state temperatures (Figure E and Table C below). This indicates the importance of the fit between the film and the device for effective optical performance and thermal management by the radiative cooling materials.

In this study, the LED chip integrated with the silicone lens was employed to characterize the performance of the metamaterials. Specifically, the silicone lens with a radius of 1.5 mm and a height of 1 mm was integrated on the LED chip. When the planar metamaterial film is directly attached on the LED chip without the silicone lens, its planar shape exhibits different light distribution from the hemispherical shape [#9-3], and the absence of the silicone layer leads to the slight sacrifice in the anti-reflection effects generated by the graded refractive index between silicone (~1.52) and c-PI (~1.50). Nonetheless, the absence of the silicone lens (thermal conductivity of silicone as 0.2~0.4 W/m·K [#9-4]) can considerably decrease thermal resistance of heat conduction, enhancing the radiative cooling effects [#9-2]. **We have added the discussions with the results to our manuscript.**

Figure E. Optical images of (Left) the LED chip covered with the silicone lens and (Right) the 50 μm thick metamaterial film integrated on top of the LED chip covered by the silicone lens. (I) bare LED chip covered by the silicone lens, (II) the film attachment with a full adhesive layer (c-PI), (III) the integration by direct coating of the mixture solution, and (IV) the film placement with the air void between the film and the device as a control sample. (Left) scale bar is 1 cm. (Right) All the scale bars are 1 mm.

Table C. Performance of commercial LED with metamaterial in terms of integration methods. The thickness of all the metamaterial samples was ~50 μm .

Sample	Integration method	Light output power @ 350 mA	Steady-state temperature	
			Dark condition	Illuminated condition
Metamaterial on LED	Film attachment with an adhesive layer	3,106 mW	37.8 °C	57.5 °C
	Integration by direct coating	3,021 mW	36.2 °C	57.3 °C
	Film placement with the air void between the film and the device	2,549 mW	40.2 °C	59.1 °C
c-PI on LED	Film attachment with an adhesive layer	2,703 mW	41.6 °C	64.8 °C
LED	-	2,628 mW	40.8 °C	64.2 °C

Reference

[#9-1] Mandal J. *et al.* Hierarchically porous polymer coatings for highly efficient passive daytime radiative cooling. *Science* **362**, 315-319 (2018).

[#9-2] Li P. *et al.* Thermo-Optically Designed Scalable Photonic Films with High Thermal Conductivity for Subambient and Above-Ambient Radiative Cooling. *Advanced Functional Materials* **32**, 2109542 (2022).

[#9-3] Chen J. *et al.* LED revolution: fundamentals and prospects for UV disinfection applications. *Environmental Science: Water Research & Technology* **3**, 188-202 (2017).

[#9-4] Mu Q. *et al.* Thermal conductivity of silicone rubber filled with ZnO. *Polymer Composites* **28**, 125-130 (2007).

Our modification to the manuscript

To clarify this, we have added Figure E and Table C to the manuscript as **Supplementary Figure 13 and Supplementary Table 4**, respectively.

Also, we have revised the manuscript on line 21 of page 14, “The thickness of the adhesive layer plays an important role to minimize thermal resistance of heat conduction for effective radiative cooling¹⁵. We further investigated the integration method of the metamaterial on the LED chip. Instead of the metamaterial film attachment by using an adhesive layer, the metamaterial layer with the resulting thickness of ~50 μm was directly integrated by using spin coating of the mixture solution on the LED chip covered by the silicone lens. The light output power and the steady-state temperature of the LED chips attached with the metamaterial film by the adhesive layer were compared with those of the LED chips integrated with the metamaterial by direct coating method. As a result, they exhibit the similar light output power (within 3%) and steady-state temperatures (within 5% in dark condition and within 1% in illuminated condition) (Supplementary Figure 13 and Supplementary Table 4), confirming that the adhesive layer do not significantly affect the thermal performance of the metamaterials. The minimized air void between the metamaterial and the device is not only crucial to achieve visible transparency due to the minimized reflection by the light scattering³², but also important to secure effective suppression of the temperature rise of LED chips, due to the minimized thermal resistance of heat conduction¹⁵. As a control experiment, the metamaterial film was placed onto the LED chip covered by the silicone lens with air void. Specifically, an adhesive layer was partially applied between the metamaterial film and the

silicone lens on the LED chip. The light output power and the steady-state temperature of the LED chips with the metamaterial film (air void) were characterized. As a result, the control sample exhibits the considerably decreased light output power and the increased steady-state temperatures (Supplementary Figure 13 and Supplementary Table 4). This indicates the importance of the interface between the metamaterial and the device for effective thermal management by the radiative cooling materials. In our study, the LED chip integrated with the silicone lens is employed to characterize the performance of the metamaterials. Specifically, the silicone lens with a radius of 1.5 mm and a height of 1 mm was integrated on the LED chip. When the planar metamaterial film is directly attached on the LED chip without the silicone lens, its planar shape exhibits different light distribution from the hemispherical shape⁶¹, and the absence of the silicone layer leads to the slight sacrifice in the anti-reflection effects generated by the graded refractive index between silicone (~1.52) and c-PI (~1.50). Nonetheless, the absence of the silicone lens (thermal conductivity of silicone as 0.2~0.4 W/m·K⁶²) can considerably decrease thermal resistance of heat conduction, enhancing the radiative cooling effects¹⁵.”

Further, we have revised the manuscript on line 12 of page 20, “Specifically, the liquid-state c-PI was spin-coated at 2,000 rpm for 3 min as an adhesive layer over the LED chip covered by the silicone lens, after which a c-PI film or a metamaterial film with a thickness of 50 μm was placed and pressed, followed by the curing process. In this process, the thickness of the adhesive layer was estimated as ~0.5 μm by measuring the thickness difference between the metamaterial film before the attachment and the metamaterial layer after the attachment. In addition, direct coating of the mixture solution on the LED chip covered by the silicone lens was carried out by spin-coating the mixture solution at 600 rpm for 5 min, followed by curing at 100 $^{\circ}\text{C}$ for 30 min and 150 $^{\circ}\text{C}$ for 30 min. The resulting thickness of the cured metamaterial on the LED chip was ~50 μm .”

Comment #10

11. It is suggested to supplement a table showing the details of the advantages of such metamaterial compared to those reported in the literature.

Response and change

Thank you for the reviewer’s comment. As in our response to Comment #1, we summarized the comparison of our transparent metamaterials with the previous “transparent” radiative cooling materials to show their key features and radiative cooling performance in Table A. As the reviewer suggested, we have added the table to our manuscript.

Our modification to the manuscript

To clarify this, we have added Table A to the manuscript as **Supplementary Table 1**.

Comment #11

12. In Figure 3a, the unit format of $^{\circ}\text{C}$ is incorrect.

13. On Page 11, a blank is missing in the sentence “...because the silicone lens (~0.7 in the wavelengths of 2.5-25 μm 25,48...”

Response and change

Thank you for the comments. We have corrected the typos of the manuscript.

Our modification to the manuscript

We have revised **Fig. 3a** of the manuscript as below.

Also, we have revised the manuscript on line 21 of page 13, “Despite small differences, the existence of c-PI on the LED chips slightly increases the maximum temperatures of the LED chips because the silicone lens (~0.7 in the wavelengths of 2.5-25 μm^{25,59}) on top of the commercial blue LED chip has a slightly higher emissivity with c-PI (~0.6 in the wavelengths of 2.5-25 μm).”

Reviewer #2 (Remarks to the Author):

Comment #12

Lee et al. reported a transparent radiative cooling metamaterials as a cover window of foldable and flexible displays by synthesis of embedding optically-modulating microstructures in polyimide. In my opinion, this research is meaningful because this material has certain practical applications. However, the characteristics of high solar transmittance and high thermal infrared emissivity are not rare, and it is not hard to achieve this performance. Therefore, I think this paper does not meet the high standard of novelty of Nature Communications. I would give comments listed below that I hope would be helpful for the revision.

Comments:

1. The Introduction chapter should explain this study's novelty. Although the authors said that most radiative cooling materials are highly solar-reflective, that is, opaque, many transparent radiative cooling materials have also been reported. What are the advantages of the authors' study compared to transparent cooling materials? What challenges does it solve?

Response and change

Thank you for the reviewer's comments. In this study, the transparent radiative cooling metamaterials are demonstrated for the cover window of foldable and flexible displays. Although many transparent radiative cooling materials have been reported, the previously reported transparent radiative cooling materials are not suitable as the cover window of foldable and flexible displays. For the successful implementation of the radiative cooling functionality to the cover windows of foldable and flexible displays, the radiative cooling materials need to meet the requirements of high optical transparency, suitable mechanical property (foldability), and excellent moisture impermeability. From that point of view, we compared our transparent metamaterials with the previous "transparent" radiative cooling materials in terms of their key features and radiative cooling performance in Table A [#12-1~#12-20]. Above all, the cover window materials of foldable and flexible displays should have high elastic modulus, enabling the excellent elastic recovery and rebound resilience [#12-21]. While SiO₂-based transparent radiative cooling materials are stiff and brittle for the cover windows of foldable and flexible displays, PDMS-based transparent radiative cooling materials do not meet the foldability requirement for the displays due to the considerably low elastic modulus. As our response to Comment #15, the repeated bending tests greatly generated the wrinkles on PDMS, showing that PDMS did not meet the requirement of the cover windows of foldable and flexible displays. In addition, TPX, silk fibroin and cellulose-based transparent radiative cooling materials have poor moisture impermeability.

In contrast, clear polyimide (c-PI) has been widely studied as a promising candidate due to its excellent optical and mechanical properties and moisture impermeability. However, c-PI film with a thickness of 50 μm exhibits low emissivity (~0.60) in the atmospheric window. To overcome this challenge, we propose the synthesis of transparent radiative cooling metamaterials based on c-PI. The outcome includes the greatly increased emissivity (from 0.60 to 0.95) in the atmospheric window under the visible transmission comparable to pristine c-PI, while further enhancing mechanical property (2.2 times higher elastic modulus) and moisture impermeability (0.6 times lower water vapor transmission rates) under the identical thickness of 50 μm. The outdoor and indoor characterization reveals the considerable suppression of the temperature rise in the display devices. To the best of our knowledge, this study is first demonstration of excellent radiative cooling performance of the metamaterials which can address the requirements of the cover windows of foldable and flexible displays. We have added the discussions to our manuscript for the better clarity.

Table A. Comparison of the recent studies on the transparent radiative cooling materials.

Material	Transmission in visible wavelengths	Emissivity in atmospheric window	Solar intensity / Relative humidity	Temperature drop (control sample)	Elastic modulus	WVTR	Reference
SiO ₂ microspheres in TPX	-	>0.93	~900 W/m ² / -	-	~360 MPa* (TPX)	~775 g/m ² ·day [†] (TPX)	[#1-1]
SiO ₂ photonic crystal arrays	-	>0.90	~1000 W/m ² / -	13 °C (Si wafer)	130~188 GPa [‡] (SiO ₂)	~2·10 ⁻³ g/m ² ·day [§] (SiO ₂)	[#1-2]
SiO ₂ microspheres on glass	~84%	>0.98	800 W/m ² / ~37%	14 °C (Si wafer)	130~188 GPa [‡] (SiO ₂)	~2·10 ⁻³ g/m ² ·day [§] (SiO ₂)	[#1-3]
Grating SiO ₂ structure	~90%	~0.90	830~990 W/m ² / 30~40%	3.6 °C (Si wafer)	130~188 GPa [‡] (SiO ₂)	~2·10 ⁻³ g/m ² ·day [§] (SiO ₂)	[#1-4]
SiO ₂ /TiO ₂ multi-layer on ITO-coated PET film	~87%	~0.88	~700 W/m ² / -	9.6 °C (Roof)	130~188 GPa [‡] (SiO ₂)	~2·10 ⁻³ g/m ² ·day [§] (SiO ₂)	[#1-5]
Silk fibroin	~91%	~0.88	~900 W/m ² / -	5.1 °C (Solar panel)	~1 GPa** (Silk fibroin)	2,000 g/m ² ·day ^{††} (Silk fibroin)	[#1-6]
PDMS on ITO-coated window	~94%	~0.90	~630 W/m ² / ~13%	7 °C (Bare window)	~2.61 MPa ^{‡‡} (PDMS)	708 g/m ² ·day ^{§§} (PDMS)	[#1-7]
n-hexadecane-infiltrated SiO ₂ aerogel microparticles in PDMS	~91%	~0.98	~920 W/m ² / 30~45%	7.7 °C (Solar cell)	~2.61 MPa ^{‡‡} (PDMS)	708 g/m ² ·day ^{§§} (PDMS)	[#1-8]
Microstructured PDMS	>90%	Close to 1.0	-	4 °C (Solar cell)	~2.61 MPa ^{‡‡} (PDMS)	708 g/m ² ·day ^{§§} (PDMS)	[#1-9]
Nanofibrillated cellulose film	>90%	>0.85	~800 W/m ² / ~27%	5 °C (Si wafer)	46 GPa*** (Cellulose)	1,315 g/m ² ·day ^{†††} (Cellulose)	[#1-10]
PMMA-SiO₂ microstructures in c-PI	~85%	~0.95	~900 W/m² / 30~40%	8.3 °C (c-PI on simulated display (400W/m²))	2.51 GPa	66 g/m²·day	Our work

* Elastic modulus of TPX: [#1-11], † WVTR of TPX: [#1-12], ‡ Elastic modulus of SiO₂: [#1-13], § WVTR of SiO₂: [#1-14], ** Elastic modulus of silk fibroin: [#1-15], †† WVTR of silk fibroin: [#1-16], ‡‡ Elastic modulus of PDMS: [#1-17], §§ WVTR of PDMS: [#1-18], *** Elastic modulus of cellulose: [#1-19], ††† WVTR of cellulose: [#1-20]

Reference

[#12-1~#12-20] Science 355 1062-1066 (2017); PNAS 112 40 12282-12287 (2015); Small 15 1905290 (2019); Renewable Energy 191 662-668 (2022); Light Sci. Appl. 11 122 (2022); Adv. Funct. Mater. 33 2301924 (2023); Cell Rep. Phys. Sci. 100231 (2020); Adv. Funct. Mater. 32 2105882 (2022); Solar energy 225 245-251 (2021); Cellulose 28 9383-9393 (2021); Membranes 10 9 (2020); Materials 10 7 821 (2017); J. Micro electro mechanical Sys. 19 2 229-238 (2010); Thin Solid Films 720 138524 (2021); Adv. Funct. Mater. 28 1705291 (2018); *Materials* 12, 3340 (2019); J. Appl. Polym. Sci., 131 41050 (2014); IEEE International Conference on MEMS 397-400 (2015); Industrial Crops and Products 138 111478 (2019); Carbohydrate Polymers 111 442-448 (2014).

[#12-21] Xie J. *et al.* Highly Foldable, Super-Sensitive, and Transparent Nanocellulose/Ceramic/Polymer Cover Windows for Flexible OLED Displays. *ACS Applied Materials & Interfaces* **14**, 16658-16668 (2022).

Our modification to the manuscript

To clarify this, we have added Table A to the manuscript as **Supplementary Table 1**.

Also, we have revised the manuscript **on line 10 of page 5**, **“We compare our transparent metamaterials with the previously demonstrated transparent radiative cooling materials in terms of their key features and radiative cooling performance in Supplementary Table 1. The cover window materials of foldable and flexible displays should have high elastic modulus, enabling the excellent elastic recovery and rebound resilience³. While transparent radiative cooling materials based on SiO₂ are stiff and brittle for the cover windows of foldable displays, PDMS-based transparent radiative cooling materials do not meet the folding requirement due to the considerably low elastic modulus. TPX, Silk fibroin and cellulose-based transparent radiative cooling materials have poor moisture impermeability. To the best of our knowledge, this study is the first demonstration of excellent radiative cooling performance of the metamaterials which can address the requirements of the cover windows of foldable and flexible displays.”**

Comment #13

2. In fact, many ordinary polymer films have the characteristics of high sunlight transmission and high thermal infrared emissivity, and do not even need a special structural design, and the material reported by the author although the sunlight transmittance is still ok, but the thermal infrared emissivity is really not high, and the emissivity begins to decline significantly after 10 microns. I think the spectral properties of this material are really mediocre, and the performance may be better if the author can design an outstanding selective emissivity.

Response and change

Although the selective emissivity in the atmospheric window is preferred to realize sub-ambient radiative cooling [#13-1], the broadband radiator is desirable to suppress the temperature rise of the device which has higher temperature than ambient by emitting more heat than the incoming atmospheric radiation, thereby creating net cooling power [#13-2]. From that point of view, our metamaterial with inclusion of PMMA-SiO₂ microstructures sized with 4-10 μm is designed to achieve broadband radiation as a cover window to mitigate the temperature rise of the displays. This study focused on demonstration of transparent radiative cooling metamaterials suited to the cover window of foldable and flexible displays. To further increase the broadband radiation of our metamaterials, the chemical modification of emissive materials [#13-3] or the microstructure optimization with respect to size [#13-4] and arrangement [#13-5] can be considered as a future study.

Reference

- [#13-1] Raman AP. *et al.* Passive radiative cooling below ambient air temperature under direct sunlight. *Nature* **515**, 540-544 (2014).
- [#13-2] Hossain MM, Gu M. Radiative Cooling: Principles, Progress, and Potentials. *Advanced Science* **3**, 1500360 (2016).
- [#13-3] Li D. *et al.* Scalable and hierarchically designed polymer film as a selective thermal emitter for high-performance all-day radiative cooling. *Nature Nanotechnology* **16**, 153-158 (2021).
- [#13-4] Wang X. *et al.* Scalable Flexible Hybrid Membranes with Photonic Structures for Daytime Radiative Cooling. *Advanced Functional Materials* **30**, 1907562 (2020).
- [#13-5] Jaramillo-Fernandez J. *et al.* A Self-Assembled 2D Thermofunctional Material for Radiative Cooling. *Small* **15**, 1905290 (2019).

Our modification to the manuscript

To clarify this, we have revised the manuscript on line 3 of page 6, “Although the selective emissivity in the atmospheric window is preferred to realize sub-ambient radiative cooling²⁰, the broadband radiator is desirable to suppress the temperature rise of the device which has higher temperature than ambient by emitting more heat than the incoming atmospheric radiation, thereby creating net cooling power³³. Since SiO₂ microstructures have different extinction coefficients in terms of their sizes^{23,26}, the broad distribution of SiO₂ microstructures contributes to increased light emission at broadband wavelengths, suppressing the temperature increase^{23,26}. Hence, SiO₂ aerogel microparticles sized 4-10 μm, where SiO₂ nanospheres with a diameter of ~20 nm form a chain-like network³⁴ (Supplementary Figure 1), are employed in metamaterials.”

Also, we have revised the manuscript on line 14 of page 8, “To further increase the broadband radiation of our metamaterials, the chemical modification of emissive materials⁴¹ or the microstructure optimization with respect to size²³ and arrangement²⁵ can be considered.”

Comment #14

3. The performance shown in Figure 3 and Figure 4 is only compared with c-PI (no PMMA-SiO₂), which is not comprehensive and meaningless, just only showing the improvement of performance by increasing PMMA-SiO₂ wt%. Readers would like to see the advantages of this material in the current field. Authors can increase the performance comparison with the current commercial films and the transparent cooling films reported in the past literature.

Response and change

As our response to Comment #12, the cover window materials for foldable display applications mainly require high optical transparency, suitable mechanical property (foldability), and excellent moisture impermeability. In addition, touch sensitivity and thermal stability (manufacturing process) should be considered for diverse display applications. From that point of view, we summarized the characteristics of various pristine materials in Table B below [#14-1~#14-32]. For the commercial products, clear polyimide (c-PI) and ultra-thin glass (UTG) have been widely studied as the candidate for foldable and flexible displays [#14-33]. UTG is mainly composed of SiO₂ which has an emissivity of ~0.75 [#14-34] in the atmospheric window. Although UTG is flexible, it has weak foldability. It should be noted that for foldable displays, the cover windows are preferred to endure the bending with a radius of 1 mm without degradation [#14-35]. The performance comparison of our metamaterials with the previously

reported transparent radiative cooling materials is summarized as Table A as our response to Comment #12. We have added the discussions to our manuscript for the better clarity.

Table B. Comparison of pristine materials in terms of optical transparency, mechanical property (foldability at a bending radius of 1 mm), moisture impermeability, thermal stability, and touch sensitivity for the cover windows of foldable and flexible displays.

Pristine material	Optical transparency (Visible wavelengths)	Mechanical property (Foldability @ bending radius of 1 mm)	Moisture impermeability (g/m ² day)	Touch sensitivity (Dielectric constant @ 1×10 ⁶ Hz)	Thermal stability
Ultra-thin glass	Good (>90%) [†]	Weak [‡] (Elastic modulus ~70 GPa, Tensile strength ~3 GPa) [‡]	Good (~0.002) [§]	Good (~3.9) ^{**}	Good (T _g = ~552 °C) ^{††}
Clear polyimide	Good (~90%) [†]	Good (Elastic modulus 1~2 GPa, Tensile strength 60-100 MPa) [‡]	Good (~165) [§]	Good (3-4) ^{**}	Good (T _g > 400 °C) ^{††}
PDMS	Good (~95%) [†]	Weak (Elastic modulus 1.3~3.0 MPa, Tensile strength 3.5~7.7 MPa) [‡]	Weak (~708) [§]	Fair (~2.4) ^{**}	Weak (T _g = -127~-121 °C, Boiling point 155~200 °C) ^{††}
PET	Good (>90%) [†]	Good (Elastic modulus ~3 GPa, Tensile strength ~80 MPa) [‡]	Good (~18) [§]	Fair (~2.2) ^{**}	Weak (T _g = 67~81 °C) ^{††}
PVDF	Good (80-90%) [†]	Good (Elastic modulus 2.5-3.5 GPa, Tensile strength 40-60 MPa) [‡]	Good (~29) [§]	Weak (~0) ^{**}	Weak (T _g = -40 °C, Melting point 171~180 °C) ^{††}
TPX	Good (>90%) [†]	Weak (Elastic modulus ~360 MPa, Tensile strength ~20 MPa) [‡]	Weak (~775) [§]	Fair (~2.0) ^{**}	Weak (T _g = 20 ~ 30 °C) ^{††}
Cellulose	Good (85-90%) [†]	Good (Elastic modulus ~46 GPa, Tensile strength 2-3 GPa) [‡]	Weak (~1,315) [§]	Good (~3.0) ^{**}	Weak (T _g = ~107 °C) ^{††}

* Foldability of ultra-thin glass (a thickness of ~ 85 μm) was reported at a bending radius of 3 mm [#14-2].

[†] Optical transparency: Ultra-thin glass[#14-1], Clear polyimide[#14-6~7], PDMS[#14-9], PET[#14-14], PVDF[#14-18], TPX[#14-23], Cellulose[#14-28]

[‡] Elastic modulus, Tensile strength: Ultra-thin glass[#14-2], Clear polyimide[#14-6~7], PDMS[#14-10], PET[#14-15], PVDF[#14-19], TPX[#14-24], Cellulose[#14-29]

[§] Water vapor transmission rate: Ultra-thin glass[#14-3], Clear polyimide[#14-8], PDMS[#14-11], PET[#14-16], PVDF[#14-20], TPX[#14-25], Cellulose[#14-30]

** Dielectric constant: Ultra-thin glass[#14-4], Clear polyimide[#14-6], PDMS[#14-9], PET[#14-17], PVDF[#14-21], TPX[#14-26], Cellulose[#14-31]

†† Thermal stability: Ultra-thin glass[#14-5], Clear polyimide[#14-7], PDMS[#14-13], PET[#14-15], PVDF[#14-22], TPX[#14-27], Cellulose[#14-32]

Our modification to the manuscript

To clarify this, we have added Table A and Table B to the manuscript as **Supplementary Table 1 and Supplementary Table 3**, respectively.

Also, we have revised the manuscript on line 21 of page 10, “For the successful implementation of our strategy to the cover windows of diverse foldable and flexible displays, considering touch sensitivity and thermal stability is also desirable in addition to optical transparency, mechanical property (foldability), and moisture impermeability. From that point of view, we summarize the characteristics of various pristine materials in Supplementary Table 3. For the commercial products, clear polyimide (c-PI) and ultra thin glass (UTG) have been widely studied as the candidate for foldable and flexible displays³. UTG is mainly composed of SiO₂ which has an emissivity of ~0.75²⁹ in the atmospheric window. Although UTG is flexible, it has weak foldability. It should be noted that for foldable displays, the cover windows are preferred to endure the bending with a radius of 1 mm without degradation⁴³.”

Reference

[#14-1~#14-32] Progress in Organic Coatings 187 108162 (2024); Solar Energy Materials and Solar Cells 166 254-261 (2017); J. Micromech. Microeng. 23 075001 (2013); J. Materials Research 28 11 1490-1497 (2013); J. Non-Crystalline Solids 324 277-288 (2003); ACS Appl. Mater. Interfaces 14 14 16658-16668 (2022); Advanced Materials Interfaces 7 2000928 (2020);

Adv. Eng. Mater. 18 582-590 (2016); Adv. Funct. Mater. 32 2105882 (2022); J. Micromechanics and Microengineering 24 035017 (2014); IEEE International Conference on MEMS 397-400 (2015); Adv. Funct. Mater. 30 4 (2020); Polymers 13 7 1080 (2021); Applied Surface Science 254 3504-3508 (2008); Composites Part B: Eng. 37 399-407 (2006); Surface and Coatings Technology 206 318-324 (2011); J. Appl. Polym. Sci. 132 42508 (2015); New Biotechnology 69, 28-35 (2022); Ferroelectrics 226 169-181 (1999); Polymer 54 1679-1690 (2013); Sci. Adv. 3 e1602902 (2017); Polymers 14 4793 (2022); Science 355 1062-1066 (2017); Membranes 10 9 (2020); Materials 10 7 821 (2017); J. Mater. Sci.: Mater. Electron. 26 9396–9401 (2015); Phys. Status Solidi (c) 6 2420-2422 (2009); ACS Sustainable Chem. Eng. 10 32 10560–10569 (2022); Industrial Crops and Products 138 111478 (2019); Carbohydrate Polymers 111 442-448 (2014); Adv. Funct. Mater. 30 1904536 (2020); Cellulose 15 445–451 (2008).

[#14-33] Xie J. *et al.* Highly Foldable, Super-Sensitive, and Transparent Nanocellulose/Ceramic/Polymer Cover Windows for Flexible OLED Displays. *ACS Applied Materials & Interfaces* **14**, 16658-16668 (2022).

[#14-34] Zhu L. *et al.* Raman AP, Fan S. Radiative cooling of solar absorbers using a visibly transparent photonic crystal thermal blackbody. *Proceedings of the National Academy of Sciences* **112**, 12282-12287 (2015).

[#14-35] Jeong S.Y. *et al.*, Foldable and washable textile-based OLEDs with a multi-functional near-room-temperature encapsulation layer for smart e-textiles. *Npj Flexible Electronics* **5** 15 (2021).

Comment #15

4. I suspect that even an unmodified polymer film, such as PDMS, would have a higher solar transmittance and thermal infrared emissivity. Many polymers themselves contain rich chemical bonds that can stretch and vibrate in the transparency atmospheric window.

Response and change

As the reviewer pointed out, pristine PDMS has the higher solar transmission and thermal infrared emissivity than pristine c-PI. While the cover window materials in foldable and flexible displays require high elastic modulus of 1-2 GPa for sufficient elastic recovery and rebound resilience [#15-1], PDMS has considerably low elastic modulus (1.3~3.0 MPa) [#15-2]. We further carried out the bending cycle test [#15-3] with a bending radius of 1 mm for 10,000 bending times to compare the foldable properties of our metamaterial film with PDMS upon the repeated folding and releasing processes. The film thicknesses were identical as 50 μm . SEM images reveals no distinct morphology changes, including wrinkle formation, of the metamaterials upon the repeated folding process (Figure F(a)), while the repeated bending cycles greatly generated the wrinkles on the surface of PDMS film (Figure F(b)) which is consistent to the previous report [#15-4]. As our response to Comment #13, we summarized the characteristics of various pristine materials in Table B for the comparison of c-PI with other polymers. **We have added the discussions to our manuscript for the better clarity.**

Figure F. Scanning Electron Microscopy (SEM) images after bending cycle test with a bending radius of 1 mm for 10,000 bending times. a, metamaterial (24 wt% PMMA-SiO₂ microstructures in c-PI). b, PDMS. The sample thicknesses were identical as 50 μm.

Reference

- [#15-1] Xie J. *et al.* Highly Foldable, Super-Sensitive, and Transparent Nanocellulose/Ceramic/Polymer Cover Windows for Flexible OLED Displays. *ACS Applied Materials & Interfaces* **14**, 16658-16668 (2022).
- [#15-2] Johnston ID. *et al.* Mechanical characterization of bulk Sylgard 184 for microfluidics and microengineering. *Journal of Micromechanics and Microengineering* **24**, 035017 (2014).
- [#15-3] Keum C. *et al.* A substrateless, flexible, and water-resistant organic light-emitting diode. *Nature Communications* **11**, 6250 (2020).
- [#15-4] Mazzocchi T. *et al.* PDMS and DLC-coated unidirectional valves for artificial urinary sphincters: Opening performance after 126 days of immersion in urine. *Journal of Biomedical Materials Research Part B: Applied Biomaterials* **110**, 817-827 (2022).

Our modification to the manuscript

To clarify this, we have added Figure F and Table B to the manuscript as **Supplementary Figure 7 and Supplementary Table 3.**

Also, we have revised the manuscript on line 24 of page 9, **“Moreover, although PDMS is a representative thermal emitter, it does not provide adequate capabilities as a cover window to enable display protection from mechanical damage, owing to its low mechanical strength (Supplementary Figure 5b and 5c). The bending cycle test on the PDMS film with a bending radius of 1 mm for 10,000 bending times greatly generates the wrinkle formation on the PDMS surface (Supplementary Figure 7) which is consistent to the previous report⁴⁵.”**

Further, we have revised the manuscript on line 18 of page 10, **“Our strategy to synthesize the metamaterials with c-PI can be applied to other materials (e.g., PET, PVDF, etc.) by replacing the optically-modulating material (PMMA) with others to consider refractive index matching, low viscosity, low surface energy, and similar solubility. For the successful implementation of our strategy to the cover windows of diverse foldable and flexible displays, considering touch sensitivity and thermal stability is also desirable in addition to optical transparency, mechanical property (foldability), and moisture impermeability. From that point of view, we summarize the characteristics of various pristine materials in Supplementary Table 3.”**

Reviewer #3 (Remarks to the Author):

Comment #16

To develop a foldable and flexible cover window with optimized radiative cooling properties, the authors dispersed PMMA-infiltrated SiO₂ aerogel microparticles into clear polyimide (c-PI). Through rational design, the cover window showed improved thermal emission in the atmospheric window with secured optical transparency as well as enhanced mechanical and moisture-impermeable properties for potential applications. The radiative cooling characteristics of the window cover in indoor and outdoor conditions were experimentally investigated. Furthermore, the authors demonstrated metamaterial-integrated light-emitting diodes and displays. Overall, this work is well-organized and the results seem promising. However, several issues should be addressed to further improve this manuscript.

1. Please explain what sample (SiO₂ wt%) was used to get the images in Fig. 1b-d and the data in Fig. 1e.

Response and change

Thank you for the positive comments. All the samples in Fig. 1b-e were 24 wt% PMMA-SiO₂ microstructures. We have revised the manuscript for the better clarity.

Our modification to the manuscript

To clarify this, we have revised the caption of Figure 1, **“Figure 1. Transparent radiative cooling materials for foldable cover window...All the samples in Fig. 1 were 24 wt% PMMA-SiO₂ microstructures.”**

Comment #17

2. In the 1st paragraph of page 7, the authors claimed that an increased optical haze enables “broad applications of displays from clear visualization to anti-glare effects”.

a) The optical haze of 0.64 might not be the optimal value for screens. The image in Fig. 1c seems quite blurry. Please specifically compare the optical haze of this work with existing technologies. It encouraged to list the optical haze of commercialized products.

b) The sample with highest PMMA-SiO₂ wt% (24 wt%) has the best radiative cooling performance and the highest optical haze (0.64). Though the optical haze of 0.64 might be too high for applications, the authors used this sample (24 wt%) in almost all following experiments (Fig. 2e, Fig. 3 and Fig. 4). What would the radiative cooling performance of other samples (with lower PMMA-SiO₂ wt%) in those experiments?

Response and change

Thank you for the comments. The commercialized anti-glare films exhibit the optical haze of 0.1~0.4 [#17-1], but the anti-glare films with higher optical haze are desirable to enhance light coupling efficiency of light-emitting diode systems for displays operating in bright environments [#17-2]. Therefore, the anti-glare film with higher haze has been widely studied for commercialization. Due to those reasons, the previous studies demonstrated the anti-glare films with the optical haze of 0.5~0.9 [#17-2~#17-4]. We characterized the optical characteristics and the radiative cooling performance of the metamaterial samples from 6 wt% to 24 wt% PMMA-SiO₂ microstructures under the identical thickness of 50 μm in Figure 2a, Figure 2b, and Supplementary Figure 6. As the reviewer pointed out, the optical haze of 0.64 in the metamaterial with 24 wt% PMMA-SiO₂ microstructures may be too high for anti-glare functionality only, but it can be useful as the light diffusion films with enhanced light coupling

efficiency of light-emitting diode systems. In addition, the controllable contents of PMMA-SiO₂ microstructures can further decrease the optical haze of the metamaterials. For example, the metamaterial with 6 wt% PMMA-SiO₂ microstructures exhibit the optical haze of 0.26 (Figure 2b), while they exhibit the radiative cooling performance of 3.6 °C, while the metamaterials of 24 wt% PMMA-SiO₂ microstructures exhibit the radiative cooling performance of 6.9 °C (Supplementary Figure 6). We have added the discussion to the manuscript as follows.

Reference

[#17-1] Antiglare film and use thereof, US 6,217,176 B1 (2001); High-haze anti-glare film and high-haze anti-glare anti-reflection film, US 2023/0092571 A1 (2023).

[#17-2] Zhu H. *et al.* Biodegradable transparent substrates for flexible organic-light-emitting diodes. *Energy & Environmental Science* **6**, 2105-2111 (2013).

[#17-3] Lim Y-W. *et al.* Built-In Haze Glass-Fabric Reinforced Siloxane Hybrid Film for Efficient Organic Light-Emitting Diodes (OLEDs). *Advanced Functional Materials* **28**, 1802944 (2018).

[#17-4] Gamage S. *et al.* Transparent nanocellulose metamaterial enables controlled optical diffusion and radiative cooling. *Journal of Materials Chemistry C* **8**, 11687-11694 (2020).

Our modification to the manuscript

To clarify this, we have revised the manuscript on line 21 of page 7, “The commercialized anti-glare films exhibit the haze factor of 0.1~0.4³⁶, but the anti-glare films with higher optical haze (0.5~0.9³⁷⁻³⁹) are desirable to enhance light coupling efficiency of light-emitting diode systems for displays operating in bright environments³⁷. The capability to tune the haze factor of the metamaterials by varying the contents of PMMA-SiO₂ microstructure offers the potential for enhanced customization and optimization in the specific applications.”

Comment #18

3. In the 2nd paragraph of page 7, though adding PMMA-infiltrated SiO₂ aerogel microparticles could help boost the elastic moduli and tensile strengths of c-PI. However, it is still unclear whether this work has fulfilled all the mechanical requirements for practical applications. What are the desired elastic moduli and tensile strengths in this research field? A detailed comparison between this work and commercialized products is necessary.

Response and change

Thank you for the reviewer’s comment. For foldable displays, the cover windows are preferred to endure the bending with a radius of 1 mm without degradation [#18-1]. To accomplish the goal, the desired mechanical requirements are elastic modulus of >1 GPa, tensile strengths of >50 MPa, and elongation at break of >2.5% (film thickness of 50 μm) [#18-2~#18-4]. It should be noted that high elastic modulus enables the excellent elastic recovery and rebound resilience. Otherwise, the repeated bending cycles greatly generate the wrinkle formation on the surfaces [#18-5]. Introducing zirconia nanoparticles and cellulose nanocrystals to c-PI increased the elastic modulus from 1.43 GPa to 2.22 GPa, the tensile strengths from 66 MPa to 82 MPa, elongation at break of >4% for foldable and flexible cover windows [#18-3]. Our metamaterials increased the elastic modulus from 1.16 GPa to 2.51 GPa, the tensile strengths from 70 MPa to 80 MPa, while exhibiting elongation at break of >3.7%. We have added the discussion to the manuscript as follows.

Reference

[#18-1] Jeong S.Y. *et al.*, Foldable and washable textile-based OLEDs with a multi-functional near-room-temperature encapsulation layer for smart e-textiles. *Npj Flexible Electronics* **5** 15 (2021).

[#18-2] Chen L. *et al.* Highly Transparent and Colorless Nanocellulose/Polyimide Substrates with Enhanced Thermal and Mechanical Properties for Flexible OLED Displays. *Advanced Materials Interfaces* **7**, 2000928 (2020)

[#18-3] Xie J. *et al.* Highly Foldable, Super-Sensitive, and Transparent Nanocellulose/Ceramic/Polymer Cover Windows for Flexible OLED Displays. *ACS Applied Materials & Interfaces* **14**, 16658-16668 (2022).

[#18-4] Mao L. *et al.*, Mechanical Analyses and Structural Design Requirements for Flexible Energy Storage Devices. *Advanced Energy Materials* **7** 1700535 (2017).

[#18-5] Mazzocchi T. *et al.* PDMS and DLC-coated unidirectional valves for artificial urinary sphincters: Opening performance after 126 days of immersion in urine. *Journal of Biomedical Materials Research Part B: Applied Biomaterials* **110**, 817-827 (2022).

Our modification to the manuscript

To clarify this, we have revised the manuscript on line 21 of page 8, “For foldable displays, the cover windows are preferred to endure the bending with a radius of 1 mm without degradation⁴³. To accomplish the goal, the desired mechanical requirements are elastic modulus of >1 GPa, tensile strengths of >50 MPa, and elongation at break of >2.5% (film thickness of 50 μm)^{3,9,44}. It should be noted that high elastic modulus enables the excellent elastic recovery and rebound resilience. Otherwise, the repeated bending cycles greatly generate the wrinkle formation on the surfaces⁴⁵. Introducing zirconia nanoparticles and cellulose nanocrystals to c-PI increased the elastic modulus from 1.43 GPa to 2.22 GPa, the tensile strengths from 66 MPa to 82 MPa, elongation at break of >4% for foldable and flexible cover windows³....As a result, the addition of 24 wt% PMMA-SiO₂ microstructures to c-PI achieves 2.2 times and 1.6 times higher elastic modulus (2.51 GPa) and tensile strength (79.8 MPa) were higher than those of c-PI, while exhibiting elongation at break of >3.7% (Figure 2d and Supplementary Figure 5).”

Comment #19

4. In Fig. 2e, the authors demonstrated foldable properties of their work. However, a complete lifecycle test is critical in order to claim “excellent foldable properties upon repeated folding and releasing processes”.

Response and change

As the reviewer pointed out, we carried out the bending cycle test [#19-1] for 10,000 bending times to support the claim on the excellent foldable properties upon repeated folding and releasing processes. Specifically, 50 μm thick metamaterial film (24 wt% PMMA-SiO₂ microstructures) was bent and released with a bending radius of 1 mm as shown in Figure G. After the bending cycle test, the tested film was attached on top of the LED chip to measure light output power, confirming the consistent performance of the metamaterials upon the repeated folding process (Figure G(a)). In addition, SEM images reveal no distinct morphology changes, including wrinkle formation, of the metamaterials upon the repeated folding process (Figure G(b)), showing the excellent foldable properties under repetitive mechanical stress. We have added the discussion with the results to our manuscript.

Reference

[#19-1] Keum C. *et al.* A substrateless, flexible, and water-resistant organic light-emitting diode. *Nature Communications* **11**, 6250 (2020).

Figure G. Bending stability test results of metamaterials (24 wt% PMMA-SiO₂ microstructures) under repeated folding and releasing process. a, Light output power changes of the LED chip on which c-PI or metamaterial films are placed in terms of bending cycles. The inset schematic illustrates the experimental setup of bending test with a bending radius of 1 mm. **b**, SEM images of the metamaterial film after 10,000 bending cycles. Tested samples have the thickness of 50 μm .

Our modification to the manuscript

To clarify this, we have added Figure G to the manuscript as **Supplementary Figure 6**.

Also, we have revised the manuscript **on line 13 of page 9**, “We carried out the bending cycle test for 10,000 bending times⁵². Specifically, 50 μm thick metamaterial film (24 wt% PMMA-SiO₂ microstructures) was bent and released with a bending radius of 1 mm as shown in Supplementary Figure 6. After the bending cycle test, the tested film was attached on top of the LED chip to measure light output power, confirming the consistent performance of the metamaterials upon the repeated folding process (Supplementary Figure 6a). In addition, SEM images reveal no distinct morphology changes, including wrinkle formation, on the metamaterial surfaces upon the repeated folding process (Supplementary Figure 6b), showing the excellent foldable properties under repetitive mechanical stress.”

Comment #20

5. In Fig. 4a and 4b, the authors showed some indoor experimental results.

a) Due to the radiative heat transfer between the radiative cooler and ceiling/walls, indoor radiative cooling power is weaker compared to outdoor counterparts. It would be helpful if the net cooling power of this indoor experiment could be calculated, preferable with numerical modeling.

b) The authors used “simulated display” in the experiment. However, several key parameters (heat capacity, mass, etc.) of this simulated display might be different from those of a real display. What would be the temperature drop if a real display is used in the experiment?

c) Please clarify the reason for using AM 1.5G solar irradiation in the indoor experiments. The solar irradiation in a room is normally weaker than outdoor environments.

Response and change

Thank you for the reviewer's comment. As the reviewer pointed out, the indoor characterization accompanies the stronger re-emission by the surrounding than the outdoor characterization, which results in the weaker radiative cooling power. As the reviewer suggested in a), we estimated the cooling power of our metamaterials under indoor and outdoor environments. The net cooling power (P_{net}) can be expressed as follows [#20-1, #20-2, #20-3]:

$$P_{net}(T) = P_{rad}(T) - P_{surr}(T_{amb}) - P_{sun} + P_{non-rad}(T, T_{amb}) - P_{gen}$$

, where P_{rad} is the radiative cooling power, P_{surr} is the re-emission power from the surrounding, P_{sun} is the thermal power from solar irradiation, $P_{non-rad}$ is the non-radiative cooling power, P_{gen} is the thermal power from device heat generation, T is cooler surface temperature, and T_{amb} is ambient temperature. Specifically, the radiative cooling power (P_{rad}) is expressed as follows:

$$P_{rad}(T) = 2\pi \int_0^{\pi/2} \int_0^{\infty} I_{BB}(T, \lambda) \varepsilon_r(\lambda, \theta) \sin(\theta) \cos(\theta) d\lambda d\theta$$

, where I_{BB} represents the spectral radiance of a black body within the wavelength range of 8-13 μm at temperature T , while ε_r denotes the emissivity of metamaterial, and the angle θ is the zenith angle. Given that the experiments were conducted perpendicularly to the solar zenith, it can be assumed that the zenith angle is 0 deg. The re-emission power from the surrounding (P_{surr}) is expressed as follows:

$$P_{surr}(T_{amb}) = 2\pi \int_0^{\pi/2} \int_0^{\infty} I_{BB}(T_{amb}, \lambda) \varepsilon_r(\lambda, \theta) \varepsilon_{surr}(\lambda, \theta) \sin(\theta) \cos(\theta) d\lambda d\theta$$

Assuming T_{amb} as 25 °C, ε_{surr} , the emissivity of the surrounding, is ~0.2 in the atmospheric window (wavelengths of 8-13 μm) and ~1.0 in other infrared region for the outdoor environment, while ε_{surr} is 1.0 in all the infrared region (2.5-25 μm) for the indoor environment to consider the ceiling and walls as an enclosure. The light absorption of the metamaterial between wavelengths of 300 nm to 2.5 μm was ~3% under solar irradiance, resulting in P_{sun} for the outdoor and indoor environments as ~21 W/m² and ~23 W/m², respectively. The non-radiative cooling power ($P_{non-rad}$) is expressed as follows:

$$P_{non-rad} = h_c(T - T_{amb})$$

, where h_c represents the combined non-radiative heat transfer coefficient. The thermal power from device heat generation (P_{gen}) is 100 W/m². As a result, the net cooling power of the metamaterials in the outdoor and indoor environments are calculated as Figure H below.

In addition to address the reviewer's comment b), we further estimate the temperature response of the simulated display in comparison to the model real display by using the following equation.

$$mc \frac{dT}{dt} = |P_{rad}(T) - P_{surr}(T_{amb}) - P_{sun} + P_{non-rad}(T, T_{amb}) - P_{gen}|$$

, where m and c is mass and heat capacity of target object, respectively, and t is time. It should be noted that to examine the transient response of the experiments, the estimated temperature response was compared with the experimental results in the indoor environment. The detailed parameter values used in the simulation are listed in Table D. Consequently, the estimated temperature responses of the simulated display with c-PI and metamaterial in the indoor environment agree well with their experimental results (Figure I(a)). Based on these, the estimated temperature response of the model real display with c-PI and metamaterial in the indoor environment is shown in Figure I(b).

In Fig. 4b, the temperature response of the metamaterial film attached on top of the LED chip was measured in illuminated condition with AM 1.5G solar simulator (class AAA) under indoor environment. The effective suppression of the temperature rise of the LED chip by the metamaterial film can greatly mitigate the decrease in the light output power of the LED chip in response to the temperature rise as shown in Fig. 4d. (Our response to the reviewer's comment c)) For estimating the effects on the light output power of the displays equipped with the metamaterials operating in the outdoor environment by suppressing the temperature rise, the indoor characterization under illuminated condition with AM 1.5G solar simulator is performed to strictly quantify the light output power of the LED chip in response to the temperature rise under the controlled environmental conditions in terms of solar intensity, ambient temperature, and relative humidity. We have added the discussion with the results to our manuscript.

Reference

- [#20-1] Lee KW. *et al.* Visibly Clear Radiative Cooling Metamaterials for Enhanced Thermal Management in Solar Cells and Windows. *Advanced Functional Materials* **32**, 2105882 (2022).
 [#20-2] Zhu L, Raman AP, Fan S. Radiative cooling of solar absorbers using a visibly transparent photonic crystal thermal blackbody. *Proceedings of the National Academy of Sciences* **112**, 12282-12287 (2015).
 [#20-3] Jaramillo-Fernandez J. *et al.* A Self-Assembled 2D Thermofunctional Material for Radiative Cooling. *Small* **15**, 1905290 (2019).

Figure H. Net cooling power of transparent radiative cooling metamaterials in terms of $\Delta T (= T - T_{amb})$ and non-radiative heat exchange coefficients (h_c) for a, the outdoor and b, the indoor conditions.

Table D. The parameter values used in the simulation to estimate the temperature response of the simulated display and the model real display. It should be noted that P_{sun} is estimated under solar irradiation of AM 1.5G illumination, considering the light absorption of the displays between wavelengths of 300 nm to 2.5 μm . To estimate $P_{non-rad}$, the combined non-radiative heat transfer coefficient (h_c) was assumed as 4 $\text{W}/\text{m}^2 \cdot \text{K}$. P_{gen} was 100 W/m^2 .

Sample	Mass (g)	Heat capacity (J/kg·K)	P_{sun} (W/m ²)	Emissivity (2.5~25 μm)
c-PI on simulated display	50	850	370	0.2~0.7
Metamaterial on simulated display	50	850	380	0.7~0.9
c-PI on model real display	40	1,500	335	0.2~0.7
Metamaterial on model real display	40	1,500	340	0.7~0.9

Figure I. The estimated temperature responses of the simulated displays and the model real displays upon the integration of the metamaterials under illuminated condition. **a**, Temperature responses of the simulated displays with c-PI and metamaterial. The estimated temperature responses of the simulated displays agree well with the experimental results. **b**, Estimated temperature responses of the model real displays with c-PI and metamaterial.

Our modification to the manuscript

To clarify this, we have added Figure H, Table D, Figure I to the manuscript as Supplementary Figure 12, Supplementary Table 5, and Supplementary Figure 15, respectively.

Also, we have revised the manuscript on line 24 of page 12, “Furthermore, the theoretical estimation of our radiative cooling metamaterials^{19,25} is carried out for indoor and outdoor environments. This indicates an excellent net cooling power of ~113 W/m² and ~97 W/m² under direct solar light at ambient temperature for outdoor and indoor conditions, respectively (Supplementary Figure 12). This is comparable to the representative transparent thermal emitter of randomly dispersed silica particles in polymethylpentene with a net cooling power of ~93 W/m² 26.”

We have also revised the manuscript on line 13 of page 21, “**Estimation of net cooling power and temperature response.** The net cooling power (P_{net}) can be expressed as follows^{19,25,29}:

$$P_{net}(T) = P_{rad}(T) - P_{surr}(T_{amb}) - P_{sun} + P_{non-rad}(T, T_{amb}) - P_{gen}$$

, where P_{rad} is the radiative cooling power, P_{surr} is the re-emission power from the surrounding, P_{sun} is the thermal power from solar irradiation, $P_{non-rad}$ is the non-radiative cooling power, P_{gen} is the thermal power from device heat generation, T is cooler surface temperature, and T_{amb} is

ambient temperature. The detailed expression per each power term is followed for the previous studies^{19,25,29}. To estimate the re-emission power from the surrounding (P_{surr}), assuming T_{amb} as 25 °C, the emissivity of the surrounding is set as ~0.2 in the atmospheric window (wavelengths of 8-13 μm) and ~1.0 in other infrared region for the outdoor environment, while the emissivity of the surrounding is set as 1.0 in all the infrared region (2.5-25 μm) for the indoor environment to consider the ceiling and walls as an enclosure. The light absorption of the metamaterial between wavelengths of 300 nm to 2.5 μm was ~3% under solar irradiance, resulting in P_{sun} for the outdoor and indoor environments as ~21 W/m^2 and ~23 W/m^2 , respectively. The thermal power from device heat generation (P_{gen}) is 100 W/m^2 .

Further, the temperature response of the simulated display is estimated in comparison to the model real display by using the following equation.

$$mc \frac{dT}{dt} = |P_{rad}(T) - P_{surr}(T_{amb}) - P_{sun} + P_{non-rad}(T, T_{amb}) - P_{gen}|$$

, where m and c is mass and heat capacity of target object, respectively, and t is time. It should be noted that to examine the transient response of the experiments, the estimated temperature response was compared with the experimental results in the indoor environment. The detailed parameter values used in the simulation are listed in Supplementary Table 5.”

We have also revised the manuscript on line 5 of page 17, “Further, predicting the temperature response of the simulated display was carried out in comparison to the model real display (Supplementary Figure 15 and Supplementary Table 5), showing the potential as transparent radiative cooling cover windows for foldable and flexible displays.”

We have also revised the manuscript on line 14 of page 13, “To estimate the optical characteristics of the displays operating in the outdoor environment, the indoor characterization under illuminated condition was performed to strictly quantify the light output power of the LED chip in response to the temperature rise under the controlled environmental conditions in terms of solar intensity, ambient temperature, and relative humidity. First, the steady-state temperatures of the LED chips...”

Comment #21

6. Please specify the PMMA-SiO₂ wt% of the sample that was tested in the section of “Metamaterial-integrated light-emitting diodes and displays”.

Response and change

Thank you for the reviewer’s comment. The metamaterials with 24 wt% PMMA-SiO₂ microstructures were tested in the section of “Metamaterial-integrated light-emitting diodes and displays. We have revised the manuscript for the better clarity.

Our modification to the manuscript

To clarify this, we have revised the manuscript on line 10 of page 13, “...our transparent metamaterials (24 wt% PMMA-SiO₂ microstructures) were integrated...”

Comment #22

7. In the “Measurements of optical, mechanical, and moisture-impermeable properties” of the Method section, the authors used a UV-visible spectrometer and a FTIR spectrometer to characterize the cover window. It is encouraged to specify whether integrating spheres were used.

Response and change

We used a UV-visible spectrometer with an integrating sphere (Lambda 650S, Perkin Elmer) for UV-VIS-NIR light transmission and reflectance at wavelengths of 300-2,500 nm. However, no integrating sphere was equipped with the FTIR spectrometer for IR light measurement. We have revised the manuscript for the better clarity.

Our modification to the manuscript

To clarify this, we have revised the manuscript on line 9 of page 19, “UV-VIS-NIR light transmittance and reflectance at wavelengths of 300-2,500 nm were measured using a UV-visible spectrometer with an integrating sphere (Lambda 650S, Perkin Elmer), and IR light transmittance and reflectance in the wavelengths of 2.5-25 μm region were measured using a Fourier transform infrared spectrometer (Nicolet 6700, Thermo Scientific).”

Comment #23

8. In the “Indoor characterization of metamaterial-integrated devices” of the Method section, the authors used liquid-state c-PI as a thin adhesive layer. Since thickness is a key factor in this work, please provide the thickness of the c-PI adhesive layer. What is the thermal influence of this adhesive layer?

Response and change

As the reviewer pointed out, the thickness of the adhesive layer plays an important role to secure effective suppression of the temperature rise of LED chips, due to the minimized thermal resistance of heat conduction [#23-1]. In our study, the thickness of the adhesive layer was estimated as $\sim 0.5 \mu\text{m}$ by measuring the thickness difference between the metamaterial film before the attachment and the metamaterial layer after the attachment. We further investigate the thermal influence of the current adhesive layer by monitoring the performance of the metamaterial layer integrated on the LED chips. Specifically, the samples were prepared by directly spin coating of the mixture solution on the LED chip at 600 rpm for 5 min, followed by curing at 100 $^{\circ}\text{C}$ for 30 min and 150 $^{\circ}\text{C}$ for 30 min. The resulting thickness of the cured metamaterial layer on the LED chip is $\sim 50 \mu\text{m}$. The light output power and the steady-state temperature of the LED chips attached with the metamaterial film by the adhesive layer were compared with those of the LED chips integrated with the metamaterial by direct coating method. As a result, they exhibited the similar light output power (within 3%) and steady-state temperatures (within 5% in dark condition and within 1% in illuminated condition) (Figure E and Table C below), confirming that the current adhesive layer do not significantly affect the thermal performance of the metamaterials. As a control experiment, the metamaterial film was placed onto the LED chip covered by the silicone lens with air void. Specifically, an adhesive layer was partially applied between the metamaterial film and the silicone lens on the LED chip. The light output power and the steady-state temperature of the LED chips with the metamaterial film (air void) were characterized. As a result, the control sample exhibited the considerably decreased light output power and the increased steady-state temperatures (Figure E and Table C below). We have added the discussion with the results to our manuscript for the better clarity.

Reference

[#23-1] Li P. *et al.* Thermo-Optically Designed Scalable Photonic Films with High Thermal Conductivity for Subambient and Above-Ambient Radiative Cooling. *Advanced Functional Materials* **32**, 2109542 (2022).

Figure E. Optical images of (Left) the LED chip covered with the silicone lens and (Right) the 50 μm thick metamaterial film integrated on top of the LED chip covered by the silicone lens. (I) bare LED chip covered by the silicone lens, (II) the film attachment with a full adhesive layer (c-PI), (III) the integration by direct coating of the mixture solution, and (IV) the film placement with the air void between the film and the device as a control sample. (Left) scale bar is 1 cm. (Right) All the scale bars are 1 mm.

Table C. Performance of commercial LED with metamaterial in terms of integration methods. The thickness of all the metamaterial samples was $\sim 50 \mu\text{m}$.

Sample	Integration method	Light output power @ 350 mA	Steady-state temperature	
			Dark condition	Illuminated condition
Metamaterial on LED	Film attachment with an adhesive layer	3,106 mW	37.8 $^{\circ}\text{C}$	57.5 $^{\circ}\text{C}$
	Integration by direct coating	3,021 mW	36.2 $^{\circ}\text{C}$	57.3 $^{\circ}\text{C}$
	Film placement with the air void between the film and the device	2,549 mW	40.2 $^{\circ}\text{C}$	59.1 $^{\circ}\text{C}$
c-PI on LED	Film attachment with an adhesive layer	2,703 mW	41.6 $^{\circ}\text{C}$	64.8 $^{\circ}\text{C}$
LED	-	2,628 mW	40.8 $^{\circ}\text{C}$	64.2 $^{\circ}\text{C}$

Our modification to the manuscript

To clarify this, we have added Figure E and Table C to the manuscript as **Supplementary Figure 13 and Supplementary Table 4**, respectively.

Also, we have revised the manuscript on line 21 of page 14, **“The thickness of the adhesive layer plays an important role to minimize thermal resistance of heat conduction for effective radiative cooling¹⁵. We further investigated the integration method of the metamaterial on the LED chip. Instead of the metamaterial film attachment by using an adhesive layer, the metamaterial layer with the resulting thickness of $\sim 50 \mu\text{m}$ was directly integrated by using spin coating of the mixture solution on the LED chip covered by the silicone lens. The light output power and the steady-state temperature of the LED chips attached with**

the metamaterial film by the adhesive layer were compared with those of the LED chips integrated with the metamaterial by direct coating method. As a result, they exhibit the similar light output power (within 3%) and steady-state temperatures (within 5% in dark condition and within 1% in illuminated condition) (Supplementary Figure 13 and Supplementary Table 4), confirming that the adhesive layer do not significantly affect the thermal performance of the metamaterials...As a control experiment, the metamaterial film was placed onto the LED chip covered by the silicone lens with air void. Specifically, an adhesive layer was partially applied between the metamaterial film and the silicone lens on the LED chip. The light output power and the steady-state temperature of the LED chips with the metamaterial film (air void) were characterized. As a result, the control sample exhibits the considerably decreased light output power and the increased steady-state temperatures (Supplementary Figure 13 and Supplementary Table 4). This indicates the importance of the interface between the metamaterial and the device for effective thermal management by the radiative cooling materials.”

Further, we have revised the manuscript on line 12 of page 20, “Specifically, the liquid-state c-PI was spin-coated at 2,000 rpm for 3 min as an adhesive layer over the LED chip covered by the silicone lens, after which a c-PI film or a metamaterial film with a thickness of 50 μm was placed and pressed, followed by the curing process. In this process, the thickness of the adhesive layer was estimated as $\sim 0.5 \mu\text{m}$ by measuring the thickness difference between the metamaterial film before the attachment and the metamaterial layer after the attachment. In addition, direct coating of the mixture solution on the LED chip covered by the silicone lens was carried out by spin-coating the mixture solution at 600 rpm for 5 min, followed by curing at 100 $^{\circ}\text{C}$ for 30 min and 150 $^{\circ}\text{C}$ for 30 min. The resulting thickness of the cured metamaterial on the LED chip was $\sim 50 \mu\text{m}$.”

Comment #24

Several minor comments include:

1. The arrow in Fig. 2a is pointing to the wrong direction.
2. In Fig. 3b, there is a typo in the title of right Y-axis.

Response and change

Thank you for the comments. We have corrected the typos of the manuscript.

Our modification to the manuscript

To clarify this, we have revised Fig. 2a and Fig. 3b of the manuscript as below.

Fig. 2a

Fig. 3b

REVIEWER COMMENTS

Reviewer #1 (Remarks to the Author):

The authors have addressed all of my concerns. I recommend publication of the manuscript.

Reviewer #2 (Remarks to the Author):

I am glad to see the new round of revision by the authors, and the quality of the article has indeed been further improved. I have no questions about the data and completeness of the paper. In the authors' responses to my comments, I have a deep understanding of the authors' efforts to promote foldable transparent radiative cooling materials, and I understand that compared to other transparent radiative cooling materials, c-PI-based materials have certain advantages in terms of elastic modulus, mechanical property and moisture impermeability, but this is not great progress in science and application. Material design is not unusual, more like a combination of past radiative cooling materials. Of course, in fact, many research on radiative cooling today are facing this problem, which is the inevitable result of the development of the field to a certain extent. Therefore, I still think this paper is more suitable for journals specializing in materials.

Reviewer #3 (Remarks to the Author):

The authors have addressed most of the comments with new information and references. However, further discussion is required about the authors' response to comment #20 on page 27.

First, in Figure H(b), according to the authors, a net cooling power of ~ 100 W/m² can be achieved with zero temperature difference for the indoor conditions, which is suspicious. Specifically speaking, considering an indoor environment where all objects are 25 degrees Celsius, the ~ 100 W/m² net cooling power seems rather weird. If we further take the $P_{\text{gen}} = 100$ W/m² and $P_{\text{sun}} = 23$ W/m² into consideration, the cooler provides > 223 W/m² radiative cooling power in an indoor thermal equilibrium environment without the natural heat sink of the sky?

Looking into the equations by authors on page 27, I have two more questions:

- (1) The authors quantitatively defined ϵ_{surr} twice but fail to define ϵ_r , which seems to be a typo. Please correct that.
- (2) In the equation of re-emission power from the surrounding (P_{surr}), the authors only considered the direct re-emission from the wall to the cooler. However, even if the walls have an emissivity of 0.2 and a reflectivity of 0.8. due to the closed indoor environment, multiple times of reflectance for the thermal infrared are expected, so the ϵ_{surr} would practically be close to unity and thus P_{surr} could be much higher than the authors' calculated value.

Secondly, in the response to the comment b) on page 27, the authors calculated $mc \cdot (dT/dt)$ with absolute value signs. Please elucidate it.

Reviewer #1 (Remarks to the Author):

Comment #1

The authors have addressed all of my concerns. I recommend publication of the manuscript.

Response

Thank you for the reviewer's comments.

Reviewer #2 (Remarks to the Author):

Comment #2

I am glad to see the new round of revision by the authors, and the quality of the article has indeed been further improved. I have no questions about the data and completeness of the paper. In the authors' responses to my comments, I have a deep understanding of the authors' efforts to promote foldable transparent radiative cooling materials, and I understand that compared to other transparent radiative cooling materials, c-PI-based materials have certain advantages in terms of elastic modulus, mechanical property and moisture impermeability, but this is not great progress in science and application. Material design is not unusual, more like a combination of past radiative cooling materials. Of course, in fact, many research on radiative cooling today are facing this problem, which is the inevitable result of the development of the field to a certain extent. Therefore, I still think this paper is more suitable for journals specializing in materials.

Response and change

Thank you for the reviewer's comments. This study is first demonstration to present a rational approach to develop a new cover window of foldable and flexible displays with transparent radiative cooling metamaterials for the successful implementation of radiative cooling technology to wide applications. We think that the innovative design and in-depth characterization of the transparent radiative cooling metamaterials hold substantial potential to revolutionize the field of thermal management for electronic devices and displays. From that point of view, we believe that this multidisciplinary study can represent important advances in broad aspects from material design and fabrication to thermal management of electronic applications.

Reviewer #3 (Remarks to the Author):

Comment #3

The authors have addressed most of the comments with new information and references. However, further discussion is required about the authors' response to comment #20 on page 27.

First, in Figure H(b), according to the authors, a net cooling power of ~ 100 W/m² can be achieved with zero temperature difference for the indoor conditions, which is suspicious. Specifically speaking, considering an indoor environment where all objects are 25 degrees Celsius, the ~ 100 W/m² net cooling power seems rather weird. If we further take the $P_{gen} = 100$ W/m² and $P_{sun} = 23$ W/m² into consideration, the cooler provides > 223 W/m² radiative cooling power in an indoor thermal equilibrium environment without the natural heat sink of the sky?

Looking into the equations by authors on page 27, I have two more questions:

- (1) The authors quantitatively defined ϵ_{ps_surr} twice but fail to define ϵ_{ps_r} , which seems to be a typo. Please correct that.
- (2) In the equation of re-emission power from the surrounding (P_{surr}), the authors only considered the direct re-emission from the wall to the cooler. However, even if the walls have an emissivity of 0.2 and a reflectivity of 0.8. due to the closed indoor environment, multiple times of reflectance for the thermal infrared are expected, so the ϵ_{ps_surr} would practically be close to unity and thus P_{surr} could be much higher than the authors' calculated value.

Response and change

Thank you for the reviewer's comments.

- (1) we thoroughly checked the calculations, and we found the there was a mistake in calculating the net cooling power of transparent radiative cooling. The corrected results for the net cooling power of the metamaterials in the outdoor and indoor environments are shown as Figure A below. It should be noted that the thermal power from device heat generation (P_{gen}) as 100 W/m² was used to estimate the temperature response of the simulated display, while P_{gen} was 0 W/m² to estimate the net cooling power of the metamaterial film alone and to compare it with the previous reports [1, 2] in Figure A.
- (2) in the net cooling power calculation, the definition of ϵ_r was added to express the equation of the re-emission power from the surrounding (P_{surr}), while the equation was double-checked in comparison to the previous report [1~5].
- (3) to estimate P_{surr} , the emissivity of the surrounding was set as 1.0 in all the infrared region (2.5-25 μ m) for the indoor environment to consider the ceiling and walls as an enclosure. For the outdoor condition, the emissivity of the surrounding was set as ~ 0.2 in the atmospheric window (wavelengths of 8-13 μ m) and ~ 1.0 in other infrared region.

Accordingly, we have corrected and revised our manuscript.

Figure A. Net cooling power of transparent radiative cooling metamaterials in terms of $\Delta T (=T-T_{amb})$ and non-radiative heat exchange coefficients (h_c) for a, the outdoor and b, the indoor conditions.

Our modification to the manuscript

To clarify this, we have replaced Supplementary Figure 12 with Figure A.

Also, we have revised the manuscript on line 24 of page 12, “Furthermore, the theoretical estimation of our radiative cooling metamaterials^{19,25} is carried out for indoor and outdoor environments (Supplementary Figure 12). For indoor condition, the re-emission power from the surrounding should be high due to the existence of the ceiling and walls. When the emissivity of the surrounding is set as 1.0 in all the infrared region (2.5–25 μm) for the indoor environment to consider those as an enclosure, the net cooling power of the metamaterial film should be $<0 \text{ W/m}^2$ under direct solar light at ambient temperature. Nonetheless, the increased temperature difference between the metamaterial and the surrounding greatly increases the net cooling power of the metamaterial, indicating the effective suppression of the temperature rise of the target devices by integrating the metamaterial film. In outdoor environments, the theoretical estimation indicates the excellent net cooling power of $\sim 109 \text{ W/m}^2$ under direct solar light at ambient temperature. This is comparable to the representative transparent thermal emitter of randomly dispersed silica particles in polymethylpentene with a net cooling power of $\sim 93 \text{ W/m}^2$ ²⁶.”

We have also revised the manuscript on line 19 of page 21, “**Estimation of net cooling power and temperature response.** The net cooling power (P_{net}) can be expressed as follows^{19,25,29}:

$$P_{net}(T) = P_{rad}(T) - P_{surr}(T_{amb}) - P_{sun} + P_{non-rad}(T, T_{amb}) - P_{gen}$$

, where P_{rad} is the radiative cooling power, P_{surr} is the re-emission power from the surrounding, P_{sun} is the thermal power from solar irradiation, $P_{non-rad}$ is the non-radiative cooling power, P_{gen} is the thermal power from device heat generation, T is cooler surface temperature, and T_{amb} is ambient temperature. Specifically, the radiative cooling power (P_{rad}) is expressed as follows:

$$P_{rad}(T) = 2\pi \int_0^{\pi/2} \int_0^{\infty} I_B(T, \lambda) \varepsilon_r(\lambda, \theta) \sin(\theta) \cos(\theta) d\lambda d\theta$$

, where I_B represents the spectral radiance of a black body within the wavelength range of 2.5–25 μm at temperature T , while ε_r denotes the metamaterial emissivity, and the angle θ is the

zenith angle. Given that the experiments were conducted perpendicularly to the solar zenith, the zenith angle as 0 deg is assumed. The re-emission power from the surrounding (P_{surr}) is expressed as follows:

$$P_{surr}(T_{amb}) = 2\pi \int_0^{\pi/2} \int_0^{\infty} I_B(T_{amb}, \lambda) \varepsilon_r(\lambda, \theta) \varepsilon_{surr}(\lambda, \theta) \sin(\theta) \cos(\theta) d\lambda d\theta$$

ε_r indicates the emissivity of metamaterial. Assuming T_{amb} as 25 °C, for the outdoor environment, ε_{surr} (the surrounding emissivity) is ~0.2 in the atmospheric window (wavelengths of 8-13 μm) and ~1.0 in other infrared region, and for the indoor environment, ε_{surr} is 1.0 in all the infrared region (2.5-25 μm) to consider the ceiling and walls as an enclosure. The light absorption of the metamaterial between wavelengths of 300 nm to 2.5 μm was ~3% under solar irradiance, resulting in P_{sun} for the outdoor and indoor environments as ~21 W/m^2 and ~23 W/m^2 , respectively. The non-radiative cooling power ($P_{non-rad}$) is expressed as follows:

$$P_{non-rad} = h_c(T - T_{amb})$$

, where h_c represents the combined non-radiative heat transfer coefficient. The thermal power from device heat generation (P_{gen}) is 0 W/m^2 to estimate the net cooling power of transparent radiative cooling metamaterial film alone.”

Reference

- [1] Chen Y-H, *et al.* Eco-Friendly Transparent Silk Fibroin Radiative Cooling Film for Thermal Management of Optoelectronics. *Advanced Functional Materials* **33**, 2301924 (2023).
- [2] Lee KW. *et al.* Visibly Clear Radiative Cooling Metamaterials for Enhanced Thermal Management in Solar Cells and Windows. *Advanced Functional Materials* **32**, 2105882 (2022).
- [3] Zhu L, Raman AP, Fan S. Radiative cooling of solar absorbers using a visibly transparent photonic crystal thermal blackbody. *Proceedings of the National Academy of Sciences* **112**, 12282-12287 (2015).
- [4] Jaramillo-Fernandez J. *et al.* A Self-Assembled 2D Thermofunctional Material for Radiative Cooling. *Small* **15**, 1905290 (2019).
- [5] Raman AP, Anoma MA, Zhu L, Rephaeli E, Fan S. Passive radiative cooling below ambient air temperature under direct sunlight. *Nature* **515**, 540-544 (2014).

Comment #4

Secondly, in the response to the comment b) on page 27, the authors calculated $mc \cdot (dT/dt)$ with absolute value signs. Please elucidate it.

Response and change

In this study, the metamaterial suppresses the temperature rise of the display. While calculating the net cooling power in our response to Comment #3 was designated to the metamaterial film itself, setting the equation to express the temperature response of the display integrated with the metamaterial required the minus sign to the net cooling power term, thereby adding the absolute value sign. For the better clarity, we have re-expressed the equation to estimate the temperature response of the display integrated with the metamaterial as follows.

$$mc \frac{dT}{dt} = -P_{net}(T) = -(P_{rad}(T) - P_{surr}(T_{amb}) - P_{sun} + P_{non-rad}(T, T_{amb}) - P_{gen})$$

, where m and c is mass per unit area and heat capacity of target object, respectively, and t is time. It should be noted that P_{gen} was 100 W/m^2 in estimating the temperature response of the simulated display and model real display. **We have revised the equation of our manuscript for the better clarity.** In addition, the temperature responses of the simulated display with c-PI and with the metamaterial in the indoor environments showed that the experimental results agreed

well with the calculation results under the condition of the surrounding emissivity of ~0.95 and the combined non-radiative heat transfer coefficient of ~5 W/m²·K. Accordingly, we have revised our manuscript for the better clarity after we thoroughly double checked the calculation results and the descriptions.

Our modification to the manuscript

To clarify this, we have revised the manuscript on line 19 of page 22, “Further, the temperature response of the simulated display is estimated in comparison to the model real display by using the following equation.

$$mc \frac{dT}{dt} = -P_{net}(T) = -(P_{rad}(T) - P_{surr}(T_{amb}) - P_{sun} + P_{non-rad}(T, T_{amb}) - P_{gen})$$

, where m and c is mass per unit area and heat capacity of target object, respectively, and t is time. It should be noted that P_{gen} is 100 W/m² in estimating the temperature response of the simulated display and model real display. It should be noted that while calculating the net cooling power is designated to the metamaterial film itself, setting the equation to express the temperature response of the display integrated with the film requires the minus sign to the net cooling power term. It should be also noted that to examine the transient response of the experiments, the estimated temperature response was compared with the experimental results in the indoor environment. The detailed parameter values used in the simulation are listed in Supplementary Table 5.”

Also, we have revised Supplementary Table 5 as below.

Supplementary Table 5. The parameter values used in the simulation to estimate the temperature response of the simulated display and the model real display in indoor environments. It should be noted that P_{sun} is estimated under solar irradiation of AM 1.5G illumination, considering the light absorption of the displays between wavelengths of 300 nm to 2.5 μm. To estimate P_{surr} and $P_{non-rad}$, the surrounding emissivity (ϵ_{surr}) and the combined non-radiative heat transfer coefficient (h_c) was set as ~0.95 and ~5 W/m²·K, respectively. P_{gen} was 100 W/m². The area of the simulated display and the model real display was set as 16 cm².

Sample	Mass (g)	Heat capacity (J/kg·K)	P_{sun} (W/m ²)	Emissivity (2.5~25 μm)
c-PI on simulated display	50	850	370	0.2~0.7
Metamaterial on simulated display	50	850	380	0.7~0.9
c-PI on model real display	40	1,500	335	0.2~0.7
Metamaterial on model real display	40	1,500	340	0.7~0.9

Also, we have revised Supplementary Figure 15 as below.

Supplementary Figure 15. The estimated temperature responses of the simulated displays and the model real displays upon the integration of the metamaterials under illuminated condition in indoor environments. a, Temperature responses of the simulated displays with c-PI and metamaterial. The estimated temperature responses of the simulated displays agree well with the experimental results. **b,** Estimated temperature responses of the model real displays with c-PI and metamaterial.

REVIEWERS' COMMENTS

Reviewer #3 (Remarks to the Author):

The authors have corrected their equations and I have no more comments.